# SiMO: Single-Modality-Operable Multimodal Collaborative Perception

**Jiageng Wen**[1][*] **Shengjie Zhao**[2][†] **Bing Li**[2][‡] **Jiafeng Huang**[3], **Kenan Ye**[2], **Hao Deng**[2]
[1]Shanghai Research Institute for Intelligent Autonomous Systems, Tongji University
[2]School of Computer Science and Technology, Tongji University
[3]School of Mechatronic Engineering and Automation, Shanghai University
`{jiageng_wen, shengjiezhao, lizi}@tongji.edu.cn`

## Abstract

Collaborative perception integrates multi-agent perspectives to enhance the sensing range and overcome occlusion issues. While existing multimodal approaches leverage complementary sensors to improve performance, they are highly prone to failure—especially when a key sensor like LiDAR is unavailable. The root cause is that feature fusion leads to semantic mismatches between single-modality features and the downstream modules. This paper addresses this challenge for the first time in the field of collaborative perception, introducing **Si**ngle-**M**odality-**O**perable Multimodal Collaborative Perception (**SiMO**). By adopting the proposed **L**ength-**A**daptive **M**ulti-**M**od**a**l Fusion (**LAMMA**), SiMO can adaptively handle remaining modal features during modal failures while maintaining consistency of the semantic space. Additionally, leveraging the innovative "Pretrain-Align-Fuse-RD" training strategy, SiMO addresses the issue of modality competition—generally overlooked by existing methods—ensuring the independence of each individual modality branch. Experiments demonstrate that SiMO effectively aligns multimodal features while simultaneously preserving modality-specific features, enabling it to maintain optimal performance across all individual modalities. The implementation details can be found in `https://github.com/dempsey-wen/SiMO`.

## 1 Introduction

Environmental perception tasks are essential to make correct decisions for autonomous driving and robotic investigation. Multi-agent collaborative perception (MACP) is considered an effective strategy to overcome the issues of single-agent perception, including limited perception range and object occlusion. By sharing raw sensory data or perception features (Wang et al., 2020a), each agent acquires diverse perspectives from multiple sources, thereby complementing information in occluded and remote areas. Many methods have been proposed to achieve a better and more precise collaborative perception (Liu et al., 2020b; Vadivelu et al., 2021; Liu et al., 2020a; Hu et al., 2022; Lei et al., 2022; Lu et al., 2023).

Recently, collaborative perception with modal heterogeneity is gaining increasing attention. Initially, given the superior performance of point clouds in precise localization, most of the MACP methods rely on LiDAR. Considering the challenges that a single-modal method can easily fail under certain conditions, some research (Xu et al., 2022a; Hu et al., 2023) has begun exploring camera-based collaborative perception methods as a supplement. Then, Zhao et al. (2023); Xiang et al. (2023); Lu et al. (2024); Gao et al. (2025) have further explored the application of multimodal collaborative perception to leverage the advantages of multimodal sensors. However, these methods only consider integrating multimodal information to improve accuracy, ignoring the drawback that modal missing or damage could lead to the failure of the entire system, like a series circuit with multiple components.

---

[*]The work was done in Engineering Research Center of Key Software Technologies for Smart City Perception and Planning.
[†]Corresponding author.
[‡]Corresponding author.

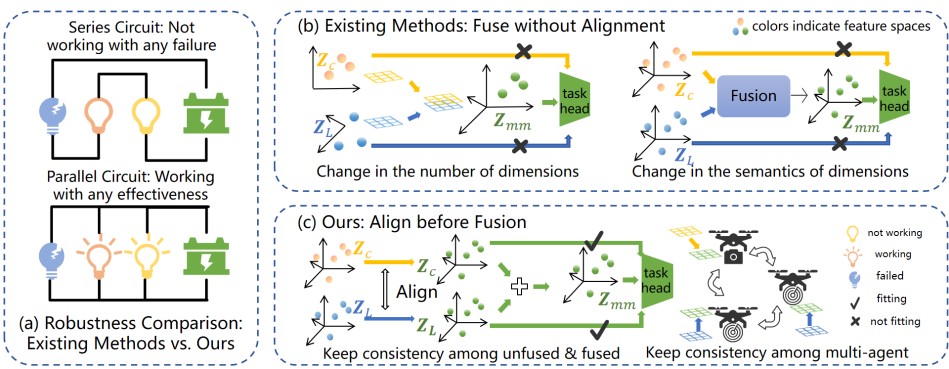

Figure 1: a) Existing methods perform like a series circuit and fail with any modal failure, while ours performs like a parallel circuit working with any effective branch. b) The common feature fusion methods cause space shifts, rendering unfused features not fitting to the downstream task heads. c) Aligning features before fusion keeps the consistency among unfused, fused and multi-agent features.

Although modal failure has been explored in single-agent scenarios (Yan et al., 2023; Ge et al., 2023; Wang et al., 2024a), MACP confronts a unique and significantly more complex challenge: Heterogeneous Modal Failure. Unlike single-agent settings that primarily require intra-agent alignment between features and local task heads, MACP demands strict inter-agent alignment. In realistic collaborative scenarios where agents suffer from different sensor failures (e.g., an ego vehicle utilizing LiDAR collaborating with a camera-only neighbor), their transmitted features must reside in a strictly unified semantic space to enable effective interaction. Existing single-agent robust methods fail to guarantee this cross-agent semantic consistency, leading to the breakdown of multi-agent fusion modules. Furthermore, the existing multimodal methods generally ignore modality competition (Huang et al., 2022), which often severely hinders multimodal joint learning. It causes models to favor easily converging branches, leaving more challenging ones inadequately trained and unable to function independently without the dominant branch. Even the dual-tower models that aim to extract multimodal features separately, like BEVFusion (Liu et al., 2023), have to rely on the only effective branch, despite the other branches could have functioned with the redundant information of multi-sensors. To our best knowledge, our work is the first to tackle the significantly more complex challenge of dynamic, heterogeneous modal failures in MACP.

We propose the **Si**ngle-**M**odality-**O**perable Multimodal Collaborative Perception (**SiMO**) method, which allows the system to work normally with any effective modality, similar to a parallel circuit working as long as one path is connected as shown in Figure 1. Compared to existing multimodal fusion methods, SiMO emphasizes aligning multimodal features before fusion and then integrating them within the same unbiased feature space. This approach not only ensures compatibility between single-modal and multimodal features for downstream modules but also maintains consistency in multi-agent features even when heterogeneous modalities fail during collaboration. We also delicately design the **L**ength-**A**daptive **M**ulti-**M**od**a**l Fusion (**LAMMA**) to handle feature fusion with modal missing. This plug-and-play fusion module can adapt to the integration of features extracted from different modalities in heterogeneous scenarios, while structurally providing a foundation for aligning the feature spaces before and after fusion. To tackle the imbalanced training of multimodal branches caused by modality competition, we introduced the "Pretrain-Align-Fuse-RD" (PAFR) training strategy, enabling the extraction of comprehensive, independent multimodal features and bolstering the efficacy of individual modalities (Figure 2(c)). We summarize our contributions as follows:

- For the first time in collaborative perception, we address multimodal perception failure caused by missing modalities with SiMO, particularly when only RGB images are available.

- We identify the inconsistency between pre- and post-fusion features as the primary catalyst for system collapse during modality failure, and propose LAMMA to accommodate varying quantities of modal features and structurally preserve semantic consistency in fusion.

- We point out that modality competition is a generally ignored obstacle to achieving the effectiveness of independent modal branches, and propose the PAFR training strategy to

solve this problem and avoid the uncertainty of balancing branch convergence as existing methods do.

- Extensive experiments show that SiMO enables each modal branch to function alone with redundant multi-modal features and keeps the SOTA level performance.

## 2 RELATED WORKS

### 2.1 MULTIMODAL COLLABORATIVE PERCEPTION

HM-ViT (Xiang et al., 2023) first explores multimodal collaborative perception with agents randomly adopting one modality of LiDAR point cloud or RGB images, achieving collaboration with modality heterogeneity. HEAL (Lu et al., 2024) further addresses the challenges of both modal and model heterogeneity in collaborative perception. Based on HEAL, STAMP (Gao et al., 2025) implements a multimodal and multi-task collaborative perception framework. BM2CP (Zhao et al., 2023) fuses point cloud and RGB image features before multi-agent collaboration, and outperforms LiDAR-based methods in 3D detection. CoBEVFusion (Qiao & Zulkernine, 2023) combines the two modalities to perform 3D detection and BEV segmentation based on BEVFusion (Liu et al., 2023). However, these existing methods cannot continue to function when LiDAR fails. Though modal failure's significant impact in multimodal perception is recognized in (Yu et al., 2023), the issue remains largely unaddressed. CMT (Yan et al., 2023) pioneered multimodal perception with single-modality operation but did not analyze the mechanisms for adapting to modal missing. MetaBEV (Ge et al., 2023) was the first to specifically tackle this problem, attributing it to feature misalignment that occurs in modal missing, a consequence of the pixel-level positional dependencies imposed by CNN+Concat fusion. UniBEV (Wang et al., 2024a) advocated identically structured feature extractors for semantic consistency, achieving more effective multimodal feature alignment than MetaBEV. Existing methods address modal failure only in single-agent, but struggle in the more complex MACP scenarios.

### 2.2 BALANCED MULTIMODAL JOINT LEARNING

Multimodal learning is typically believed to improve performance by integrating features of various modalities, outperforming single-modal approaches. However, in real-world applications, jointly trained multimodal classification networks often underperform compared to their single-modality counterparts. Wang et al. (2020b) initially investigate the phenomenon, identifying overfitting resulting from increased model capacity from multiple branches as the primary cause, and introduce Gradient Blending as a solution. Huang et al. (2022) firstly conclude the reason as modality competition and elaborate on how it hampers joint training of multimodal networks, suggesting that random initialization contributes to disparities in modal advantages. To rectify imbalanced training, Peng et al. (2022) prove that adjusting update gradients and dropout ratios for different branches during joint learning can ensure that both modal branches converge at a similar pace. Wei et al. (2025) argue that prioritizing the weaker modality for balance can limit the dominant modality's performance. Instead, they propose to assess the trainability of different modal representations based on their separability and adjust the relearning intensity accordingly. Yang et al. (2024) highlight the importance of element-wise gradient modulation, considering the varying significance of each network parameter. Current methods struggle to balance convergence rates and determine optimal modulation parameters, necessitating more deterministic approaches to address modality competition, especially in complex multimodal collaborative perception.

## 3 METHODOLOGY

### 3.1 THE ARCHITECTURE OVERVIEW

3D object detection is a prevalent challenge in MACP, so we frame the problem as multiple agents using point clouds and multi-perspective images to detect objects in 3D. To facilitate the fusion of features from various agents, it is necessary to transform these features based on their respective poses. Consequently, Bird Eye's View (BEV) feature with explicit spatial semantics has emerged as a prevalent feature type in collaborative perception. As shown in Figure 2(a), the model processes these inputs—point clouds $\mathbf{X}_L$ and RGB images $\mathbf{X}_C$—by extracting their BEV features and aligning

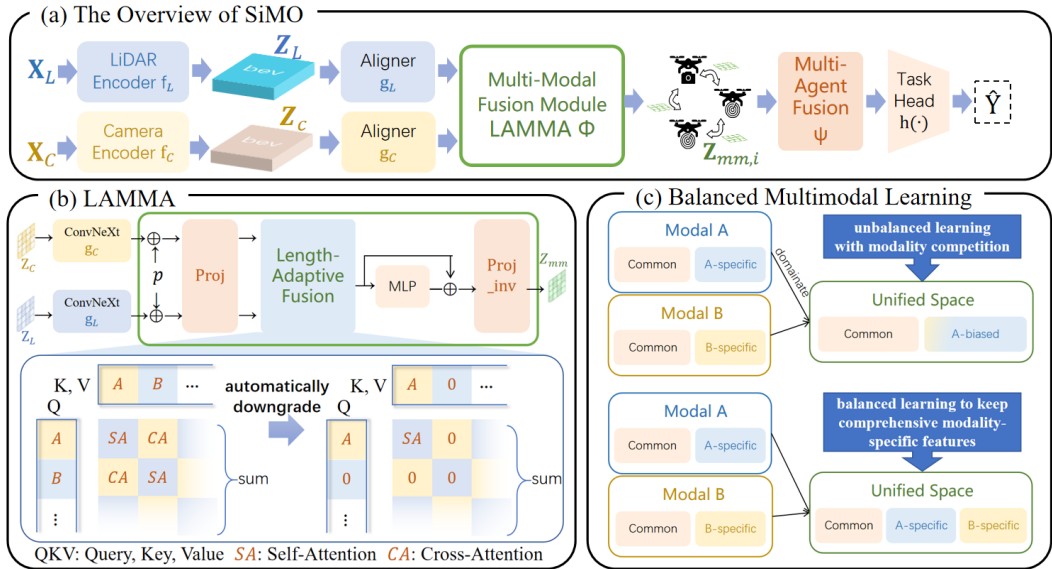

Figure 2: a) The overview of SiMO. b) LAMMA adaptively downgrades to Self-Attention fusion to keep consistent feature processing when modal failure happens. c) SiMO conducts balanced multimodal learning to keep modality-specific features for branch independence.

them to similar feature spaces to attain $\mathbf{Z}_L$ and $\mathbf{Z}_C$, followed by the creation of multimodal features $\mathbf{Z}_{mm}$ through the multimodal fusion $\Phi$. The BEV features $\mathbf{Z}_{mm,i}$ shared by various agents are then fused to form a collaborative feature, and finally input to cls, reg and dir task heads h to produce 3D bounding boxes $\hat{\mathbf{Y}}_{reg}$ and corresponding classification results $\hat{\mathbf{Y}}_{cls}$. This process is described as:

$$\mathbf{Z}_m = \mathrm{g}_m[\mathrm{f}_m(\mathbf{X}_m)],\ m = L, C \tag{1a}$$
$$\mathbf{Z}_{mm} = \Phi(\mathbf{Z}_L, \mathbf{Z}_C) \tag{1b}$$
$$\hat{\mathbf{Y}} = \mathrm{h}[\Psi(\mathbf{Z}_{mm,i})],\ i = 1, 2, 3, ... \tag{1c}$$

where $\mathbf{Z}_m$ represents the aligned BEV features, with $\mathrm{f}_m$ denoting the feature extractor and $\mathrm{g}_m$ the feature alignment module for modality $m$. $\Phi$ represents the multimodal fusion, $\Psi$ the multi-agent fusion, and h the task heads. Besides, the subscript $L$ indicates the LiDAR modality, $C$ the camera, $mm$ the fused multi-modality, and $i$ for agent order.

We adopt supervised training for multimodal joint learning. OPV2V-H (Lu et al., 2024) provides 3D bounding boxes for all vehicle objects within the perception range of multiple agents. Each bounding box label $\mathbf{Y}$ encompasses its central coordinates $(x, y, z)$, dimensions (height, width, length - $h, w, l$), orientation angle ($\theta$), and target class ($\mathbf{Y}_{cls}$). Thus, the comprehensive 3D object detection task is essentially a hybrid of a seven-value regression task and a classification task. We apply Focal Loss (Lin et al., 2017) for classification and Smooth-L1 Loss (Girshick, 2015) for regression. The total loss is as follows:

$$\mathrm{L}(\hat{\mathbf{Y}}, \mathbf{Y}) = \mathrm{L}_{Focal}(\hat{\mathbf{Y}}_{cls}, \mathbf{Y}_{cls}) + \mathrm{L}_{Smooth-L1}(\hat{\mathbf{Y}}_{reg}, \mathbf{Y}_{reg}). \tag{2}$$

## 3.2 WHY THE MODEL FAILS WITH SINGLE MODALITY?

Multimodal collaboration often uses concatenation, convolutional neural network (CNN), graph neural networks (GNN) (Scarselli et al., 2008), attention mechanisms, or Transformer (Vaswani, 2017) to blend features from various modalities. These methods either combine different modalities' features by merging their dimensionality (Liu et al., 2023; Qiao & Zulkernine, 2023), or integrate features into a new feature space (Xiang et al., 2023; Zhao et al., 2023). These fusions create a disparity between the pre- and post-fusion feature spaces, making downstream task heads designed for fused features unsuitable for unfused ones. Consequently, when a modality is absent and feature fusion is inapplicable, these methods lose their effectiveness (Figure 1(b)).

Taking these points into account, we first opt to integrate features from various modalities into a single feature space using joint learning. Within the framework of joint learning, upon processing by identical modules post-fusion, it can be inferred that the features of multiple branches reside within the same feature space. Subsequently, via addition to fuse these features, we ensure that the feature space before and after fusion remains consistent. This approach enables both fused and individual modal features to be smoothly fed to downstream task heads (Figure 1(c)).

### 3.3 LENGTH-ADAPTIVE MULTIMODAL FUSION (LAMMA)

Additive fusion hinges on the effective alignment of multimodal features. While joint multimodal learning aids this goal, as Wang et al. (2024a) highlights, a consistent feature processing architecture is crucial for semantic alignment. However, in MACP's typical heterogeneous scenarios, diverse modal feature extraction is common. Therefore, we aim to achieve consistent multimodal feature processing within the fusion module itself, avoiding additional requirements on other components. Furthermore, modal failure in collaborative settings implies a variable number of input features, and detecting and responding to this change in complex, unpredictable real-world environments is challenging. To address these issues, we propose Length-Adaptive Multimodal Fusion (LAMMA), which is structurally designed for consistent multimodal feature processing and adaptive fusion.

Extracted features originating from distinct modalities exhibit considerable semantic variations, and multimodal fusion without prior semantic unification could result in mutual interference and detract from the model's efficacy. Hence, we apply ConvNeXt (Liu et al., 2022) as $g_L$ and $g_C$ aligners to align multimodal features $Z_L$ and $Z_C$'s semantics in both channel-wise and pixel-wise. Then, all input features are added with a modality-agnostic positional embedding avoiding modality-special treatment, and projected to more compact feature spaces for length-adaptive fusion. The attention-based length-adaptive fusion robustly handles missing modalities, which is a common challenge for existing methods based on CNN, RNN or Transformer. We achieve this by concatenating queries from different modalities in parallel, allowing for simultaneous self-attention and cross-attention. The cross-attention mechanism facilitates both the transfer of complementary information and the alignment of shared features across modalities. The final integrated multimodal feature representation is then derived by additively combining these fused self- and cross-attention results. Hence, our length-adaptive fusion is implemented as:

$$\mathbf{Q} = \mathbf{W_Q}[\mathbf{Z}_A; \mathbf{Z}_B] \in \mathbb{R}^{b \times 2n \times d} \tag{3a}$$

$$\mathbf{K}_m = \mathbf{W_K}\mathbf{Z}_m \in \mathbb{R}^{b \times n \times d} \tag{3b}$$

$$\mathbf{V}_m = \mathbf{W_V}\mathbf{Z}_m \in \mathbb{R}^{b \times n \times d} \tag{3c}$$

$$\mathbf{Z}_{att\_m} = \text{MHA}(\mathbf{Q}, \mathbf{K}_m, \mathbf{V}_m) \in \mathbb{R}^{b \times 2n \times d} \tag{3d}$$

$$\mathbf{Z}_{fused\_m} = \text{Sum}(\text{Split}(\mathbf{Z}_{att\_m})) \in \mathbb{R}^{b \times n \times d} \tag{3e}$$

$$\mathbf{Z}_{mm} = \mathbf{Z}_{fused\_A} + \mathbf{Z}_{fused\_B} \in \mathbb{R}^{b \times n \times d} \tag{3f}$$

where [;] means feature concatenation, and $\mathbf{W_Q}, \mathbf{W_K}, \mathbf{W_V}$ are the linear weights for $query$, $key$ and $value$ respectively, shared by all modal features. $query$s for $A$ and $B$ are concatenated as $\mathbf{Q}$, while $key$s and $value$s are kept separated as $\mathbf{K}_m$ and $\mathbf{V}_m$ with $m = A, B$. $b, n, d$ are batch size, number of tokens and feature dimension respectively. Here $A, B$ are used to refer to different but not specific modalities, as we do not assign the order of modalities, and LAMMA should treat all modalities consistently. $\mathbf{Z}_{att\_m}$ attained with multi-head attention (MHA), are split to two parts in the size of $(b, n, d)$, of which one for self-attention fusion and another for cross-attention fusion, and then summed up in element-wise to give $\mathbf{Z}_{fused\_m}$ as enhanced representations for each input modality. Finally, $\mathbf{Z}_{fused\_A}$ and $\mathbf{Z}_{fused\_B}$ are fused with addition for $\mathbf{Z}_{mm}$ to avoid shifting the feature space. In the case of modal failure, the absent feature (e.g., $\mathbf{Z}_A$) in the query is empty, resulting in $\mathbf{Q} = [0; \mathbf{Q}_B]$. Consequently, the fusion can be considered as essentially downgrading to a self-attention module (SA) without the need to distinguish failure happening (Figure 2(b)):

$$\mathbf{Z}_{fused\_m} = \text{Softmax}(\frac{\mathbf{Q}_A\mathbf{K}_m^T}{\sqrt{d}})\mathbf{V}_m + \text{Softmax}(\frac{\mathbf{Q}_B\mathbf{K}_m^T}{\sqrt{d}})\mathbf{V}_m \tag{4a}$$

$$\mathbf{Z}_{fused\_A} = \mathbf{0}, \quad \mathbf{Z}_{fused\_B} \approx \text{SA}(\mathbf{Q}_B, \mathbf{K}_B, \mathbf{V}_B). \tag{4b}$$

Given that the model remains unchanged regardless of the presence or absence of modalities, and the input is consistently processed with exactly the same parameters, the semantic interpretation of

the fused features is supposed to remain consistent. So, no matter if a modality drops out or not, the fused features share the same semantics, ensuring optimal adaptability to the downstream task heads.

## 3.4 OVERCOMING MODALITY COMPETITION

### 3.4.1 MODALITY COMPETITION

To achieve object detection with only a single modality, it is crucial to respectively extract full features from different modalities and align them in a shared space through multimodal joint learning. However, we have observed (details in Table 11 of Section A.9) that a multimodal model trained jointly performs even worse than the best single-modality (LiDAR) model. The counter-intuitive phenomenon, termed as 'modality competition' by Huang et al. (2022), is believed to result in insufficient training of certain branches in naive joint learning. Prior studies (Wang et al., 2020b; Huang et al., 2022; Peng et al., 2022; Javaloy et al., 2022; Wei et al., 2025; Yang et al., 2024) attribute these to factors like varying optimization distances or model overfitting, with various mitigation strategies proposed. However, we believe the phenomenon stem from a more fundamental disparity in how modalities present task-relevant information.

Building on these observations and existing research, particularly the faster convergence of LiDAR models, we hypothesize that modalities inherently vary in their efficiency at providing 'task-relevant information density'. For instance, extracting 3D spatial data is far more direct from point clouds than inferring it from 2D images. We contend this fundamental difference in 'information carrying efficiency' is a primary driver of modal competition. Modalities with higher efficiency (e.g., LiDAR for 3D geometry) enable faster learning in their respective branches, which can then dominate the joint optimization process and suppress the thorough training of branches associated with lower-efficiency modalities (e.g., camera for 3D from 2D). This perspective suggests that issues like 'varying optimization distances' (Huang et al., 2022) or 'model overfitting' (Wang et al., 2020b) (potentially to the more efficient modality) can be seen as manifestations of this underlying efficiency gap. Given these intrinsic differences in 'information carrying efficiency,' we argue that modal competition is largely inevitable within traditional end-to-end joint training paradigms, where an imbalanced learning dynamic naturally arises from co-optimizing modalities with disparate ease of information extraction.

### 3.4.2 SOLUTION

Recognizing this, our research shifts from attempting to balance competition within such a framework. Instead, we aim to circumvent its negative effects through an isolated multi-branch training approach (shown in Figure 3). This strategy involves independently pre-training each modality-specific branch to develop comprehensive feature extraction capabilities before any subsequent fusion. Our goal is not to eliminate inherent modal differences but to architect a training paradigm that prevents these differences from causing detrimental competition, especially in the early learning stages, thereby maximizing each modality's potential contribution.

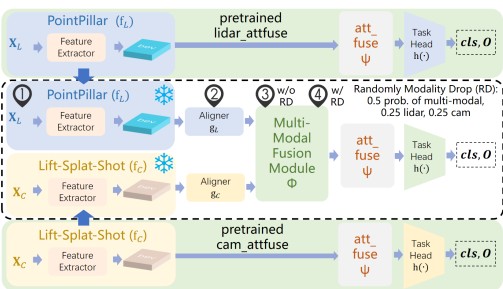

Figure 3: The training process of SiMO. (1) Load pretrained feature extractors. (2) Train each aligner with the extractor frozen. (3) Train LAMMA with two-modal input, freezing aligners and task heads (w/o RD). (4) Fine-tune LAMMA with RD to adapt modal failure.

(**Step 1**) Instead of training both branches together, we consider training each branch independently to guarantee sufficient training for both branches as a strategy with more certainty. So, feature extractors that have been pre-trained through single-modal learning will be directly adopted after convergence of both branches. Then, we maintain these pre-trained extractors in a frozen state while advancing the training of the feature alignment module, fusion module, and task-specific heads.

(**Step 2**) Given that the effective features extracted by the pre-trained feature extractors reside in different feature spaces, the subsequent step involves training an alignment module capable of aligning these features into the spaces compatible to the fusion module. We initiate the training process with the model accepting single-modal inputs (e.g. LiDAR), continuing until the model

achieves convergence. Subsequently, we freeze the trained alignment module (e.g. $g_L$), and proceed to train the other aligner (with only camera inputs) until the model converges once more. This ensures that the second aligner ($g_C$) can also effectively align its features to a LAMMA compatible space. More aligners for other modalities can be trained and frozen in the same way.

**(Step 3)** Finally, with all alignment modules frozen, we train the rest modules with multimodal inputs to achieve final convergence. All parameters of the common modules, including multimodal fusion, multi-agent fusion and task heads, are shared by all inputs without modality notations, so the common modules will be modality-independent.

Throughout the training of feature extractors and fusion modules, our approach is to prioritize a single branch each time to preclude the emergence of modality competition, then freeze the branches for the common modules training. After the convergence of common modules, the model is fine-tuned with 0.5 possibility to **R**andomly **D**rop one modal feature in LAMMA (**RD**) in order to fit the model to the case of single-modal failure **(Step 4)**.

# 4 EXPERIMENTS

## 4.1 EXPERIMENTAL SETUP

**Dataset**. We conduct our experiments on the public MACP dataset OPV2V-H (Lu et al., 2024), a heterogeneous collaborative perception dataset based on OPV2V (Xu et al., 2022c) with a supplementary of more sensor types. With the autonomous driving simulator CARLA (Dosovitskiy et al., 2017) to simulate the road environments of cities and suburbs, OPV2V collects 73 scenes with 11,464 frames in total. The whole dataset (not including culver_test set) is split into train, validate and test set with 6374, 1980 and 2170 frames respectively. Each frame contains the perception data of 2-7 Connected Automated Vehicles (CAVs), including the $800\times600$ RGB images of the front, back, right and left cameras and the point clouds of a 32-channel LiDAR. 3D bounding box annotations within the perception range of CAVs are provided for collaborative 3D object detection. In Section A.7, we also validate SiMO on more datasets.

**Model Settings**. We adopt Lift-Splat-Shot (LSS) (Philion & Fidler, 2020) for camera encoder and PointPillar (Lang et al., 2019) for LiDAR encoder to independently extract BEV features of multi-view images and point cloud, and LAMMA for multimodal fusion. The aligners to unify the semantics of two features are two independent 3-layer ConvNeXt blocks (Liu et al., 2022). As for multi-agent fusion, we choose AttFusion (Xu et al., 2022c) because the absence of trainable parameters in AttFusion facilitates the training of downstream modules shared among multiple branches. Specifically, we load the pre-trained model parameters of $opv2v\_cam\_attfuse$ and $opv2v\_lidar\_attfuse$, which both incorporate AttFusion with LSS for camera feature extraction and PointPillar for LiDAR, and then freeze the extractor to continue to train LAMMA. We also apply the Pyramid Fusion model (Lu et al., 2024)

Table 1: Quantitative 3D detection performances of models (in %).

| Method | Modality | AP@30 | AP@50 | AP@70 |
|---|---|---|---|---|
| BM2CP (Zhao et al., 2023) | L+C | 91.69 | 91.45 | 86.87 |
|  | L | 91.55 | 91.31 | 86.80 |
|  | C | 0 | 0 | 0 |
| BEVFusion (Liu et al., 2023) | L+C (RD) | 95.18 | 94.21 | 81.09 |
|  | L (RD) | 93.27 | 91.99 | 71.80 |
|  | C (RD) | 0 | 0 | 0 |
| UniBEV(Wang et al., 2024a) | L+C (RD) | 93.33 | 91.71 | 70.75 |
|  | L (RD) | 93.32 | 91.73 | 70.78 |
|  | C (RD) | 1.93 | 0 | 0 |
| AttFusion (Xu et al., 2022c) | L | 96.68 | 95.09 | 87.16 |
|  | C | 68.31 | 52.91 | 25.30 |
| SiMO (AttFusion) | L+C (w/o RD) | 96.51 | 95.26 | 85.10 |
|  | L+C (RD) | 96.10 | 94.98 | 84.81 |
|  | L (RD) | 95.11 | 94.02 | 82.35 |
|  | C (RD) | 66.53 | 49.69 | 22.59 |
| HEAL (Lu et al., 2024) (Pyramid Fusion) | L | 98.22 | 98.00 | 96.16 |
|  | C | 68.45 | 60.48 | 39.07 |
| SiMO (Pyramid Fusion) | L+C (w/o RD) | 98.38 | 98.05 | 94.89 |
|  | L+C (RD) | 98.30 | 97.94 | 94.64 |
|  | L (RD) | 97.32 | 97.07 | 94.06 |
|  | C (RD) | 80.81 | 69.63 | 44.82 |

as the main body of collaborative perception model with LAMMA to fuse multimodal features, verifying the effectiveness of LAMMA as a plug-and-play module to fit different collaborative perception frameworks. More details are in Section A.1.

**Train Settings**. Our models are trained with Adam (Kingma, 2014) optimizer and batchsize 1 on NVIDIA GeForece RTX 3090 GPUs. The perception range is 102.4*102.4 centered around the ego CAV. Average precisions at Intersection-over-Union thresholds of 0.30 (AP@30), 0.50 (AP@50), 0.70 (AP@70) are adopted to measure the model performance. Inheriting the pretrained feature extractors open-sourced by HEAL (Lu et al., 2024), we freeze the parameters of extractor to further train the aligner and task heads. LiDAR branch training starts with an initial learning rate (lr) 2e-4

and weight decay 1e-4, and reduces by 0.1 at epoch 3 until convergence. After that, camera branch is trained with the same setting except task heads frozen. Freezing extractors, aligners and task heads, fusion module LAMMA is trained with lr of 2e-5.

## 4.2 QUANTITATIVE RESULTS

**Performance Analysis**. We take BM2CP (Zhao et al., 2023), the only open-sourced MACP method with multimodal fusion, as our baseline, and we transfer BEVFusion (Liu et al., 2023), a popular SOTA multimodal method for single agent, to collaborative perception by adding Pyramid Fusion for multi-agent collaboration. To control variables and limit the scope of comparison to multimodal fusion, we replace the LiDAR BEV feature extractor of BEVFusion with PointPillar and in the same settings with SiMO. Similarly, to compare with the SOTA methods for solving modal failure in single-modal scenarios, we adopt the CNW (Channel-Normalized Weights) fusion proposed by UniBEV as the multimodal fusion, mirroring the other setup used for SiMO, and train the model with the Modality Dropout method proposed in its original paper. Moreover, we consider comparisons with the single-modal models of AttFusion and HEAL, which is the current SOTA collaborative perception method. Table 1 shows that none of BM2CP, BEVFusion and UniBEV cannot operate when LiDAR failing, while our SiMOs can effectively work with either single modal or multi modals, especially with only camera images. SiMO-AF is based on the pretrained models of AttFusion, and SiMO-PF based on Pyramid Fusion. Before fine-tuning with RD, SiMO-AF and SiMO-PF keep the upper performance limits of AttFusion and Pyramid Fusion (SOTA) respectively, which is predictable as we adopt the same feature extractors of pretrained AttFusion and Pyramid Fusion. SiMO is designed to extract the comprehensive features for both LiDAR and camera to enable each branch, causing redundancy of the common feature of LiDAR and camera. So the results suggest that LAMMA can effectively handle the redundancy of two-modal features.

*After fine-tuning with RD, both SiMO-AF and SiMO-PF achieve the adaptability to modal failure of either LiDAR or camera.* SiMOs with RD maintain the performance levels as before RD fine-tuning, without suffering a severe deterioration caused by modal failure. Moreover, SiMO-PF with camera modal exhibits a significant improvement (12.36/9.15/5.75 for AP@30/@50/@70) over Pyramid Fusion's, which suggests that Pyramid Fusion does not take fully advantage of the camera extracted features, considering their extracted features are exactly the same. While LAMMA can not only fuse the multimodal features, but also further refine the features for tasks.

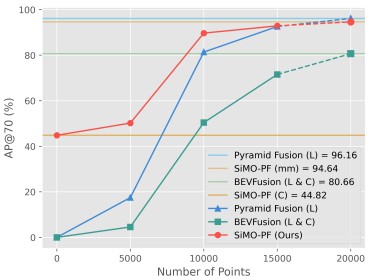

Figure 4: Comparison of SiMO-PF, BEVFusion and Pyramid Fusion (L) in varying extents of LiDAR failure.

**Robustness against Modal failure**. To further demonstrate the robustness of SiMO against LiDAR failure, we evaluate the performances of SiMO-PF and Pyramid Fusion with varying numbers of LiDAR points, simulating the different degresses of LiDAR failure. Figure 4 shows that as LiDAR fails, the performances of Pyramid Fusion with LiDAR and BEVFusion drop rapidly and eventually fail to detect, while only SiMO-PF's performance degrades in a much slower speed with the complementation of camera modality and finally converges to the level of SiMO-PF with only camera modality.

**Adaptability to Heterogeneous Modal failure**. The above comparisons happen in homogeneous modal failure settings, where either LiDAR or camera is effective for all agents, while for heterogeneous modal failure, different agents in the collaboration graph lose distinct sensors. SiMO aligns BEV features into a consistent space for effective fusion regardless of failing modalities, so it is suppose to adapt to heterogeneous failure as well. In our experiments, we simulate this using an alternating failure pattern, where agents

Table 2: Performances in homogeneous and heterogenous modal failures. L+C: both modalities, L: only LiDAR, C: only cameras, C-ego: heterogeneous failure with camera ego (C-L-C-L-...), L-ego: heterogeneous failure with LiDAR ego (L-C-L-C-...).

| METHOD | HEAL (PYRAMID FUSION) | | | | SiMO-PF | | | | |
|---|---|---|---|---|---|---|---|---|---|
| MODE | L | C | C-EGO | L-EGO | L+C | L | C | C-EGO | L-EGO |
| AP@30 | 0.98 | 0.68 | 0.85 | 0.96 | 0.98 | 0.97 | 0.81 | 0.89 | 0.97 |
| AP@50 | 0.98 | 0.60 | 0.82 | 0.96 | 0.98 | 0.97 | 0.70 | 0.85 | 0.97 |
| AP@70 | 0.96 | 0.39 | 0.72 | 0.92 | 0.95 | 0.94 | 0.45 | 0.70 | 0.91 |

sequentially lose different modalities (e.g., Modality A, then Modality B, then Modality A, and so on). Table 2 shows that compared to homogeneous failure, heterogeneous failure with LiDAR ego (the ego agent is equipped with LiDAR) results in only minimal performance decline, and models with camera ego significantly improve performances over camera-only setups. Moreover, compared to HEAL, which is designed for heterogeneous failure (collaboration among agents with heterogeneous modality), SiMO-PF achieves higher detection accuracies with both LiDAR ego and camera ego in the same heterogeneous failure settings without additional fine-tuning.

### 4.3 ABLATION STUDY

The ablation study is conducted to prove that achieving SiMO's adaptability to modal failure requires applying our learning strategy, RD fine-tuning, and LAMMA simultaneously. The comparisons between BEVFusion and SiMO-PF in Table 1 have demonstrated that even with RD fine-

Table 3: Ablation study shows the necessity of our learning strategy, RD and LAMMA to adapt to modal failure.

| OUR LEARNING STRATEGY | RD | LAMMA | CORRESPONDING EXPERIMENTS | AP@70 (L+C/L/C) | ADAPTABILITY TO MODAL FAILURE |
|---|---|---|---|---|---|
| | ✔ | ✔ | NAIVE TRAINING W/O RD NAIVE TRAINING WITH RD | 0.94/0.01/0 0.11/0/0 | ✗ |
| ✔ | | ✔ | SiMO W/O RD | 0.95/0.26/0 | ✗ |
| ✔ | ✔ | | BEVFUSION WITH RD | 0.81/0.72/0 | ✗ |
| ✔ | ✔ | ✔ | SiMO WITH RD | 0.95/0.94/**0.45** | ✔ |

tuning, the model fails to adapt to modal failure without LAMMA for multimodal feature fusion. The performance of SiMO before and after RD highlights the importance of RD in enhancing adaptability to modal failure (shown in Table 3). To further examine the impact of our joint learning method, we also trained SiMO-PF naively. As a result, regardless of whether RD fine-tuning was applied, SiMO-PF was unable to adapt to modal failure. The performance of the naively trained SiMO-PF with RD fine-tuning further confirms that without addressing modal competition through our training strategy, RD would only hinder the model. More ablation studies can be founded in Section A.9.

**The Effectiveness Verification of LAMMA.** To quantitatively validate that LAMMA successfully aligns features from different modalities into a common, semantically consistent space while preserving their modality-specific characteristics, we conducted both Procrustes analysis and t-SNE visualization. Procrustes analysis (Table 4), which measures the disparity between two feature sets after an optimal linear transformation, showed a dramatic reduction in disparity between LiDAR and camera features after processing by LAMMA (from 0.6747 to 0.0472). This indicates that the features' geometric structures became highly congruent. Furthermore, t-SNE visualizations (Figure 6) revealed that while the features naturally formed distinct clusters according to their modality (preserving modality-specific information), their intra-cluster topological structures exhibited a high degree of mirror symmetry after alignment.

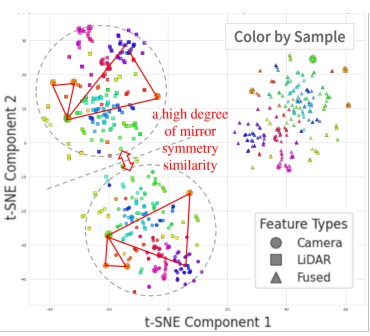

Figure 6: The t-SNE visualization for features of LAMMA.

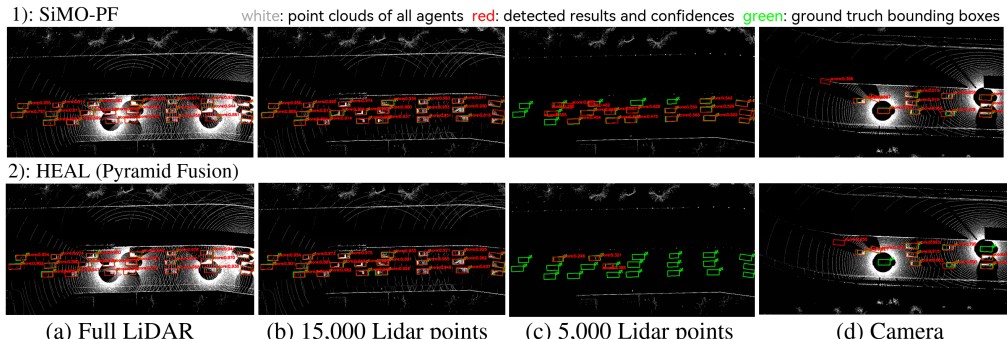

(a) Full LiDAR     (b) 15,000 Lidar points     (c) 5,000 Lidar points     (d) Camera

Figure 5: Visualization of SiMO-PF's and HEAL's (Pyramid Fusion) detection results, ground truth and collaborative point clouds. (a)(b)(c) are detection visualizations with all, 15000, 5000 LiDAR points (by occluding local point clouds), and (d) is detection visualizations with only camera images. SiMO-PF detects most objects even with 5000 LiDAR points, while HEAL misses the most.

This suggests that semantic relationships between samples are consistently encoded across modalities. These findings empirically confirm that LAMMA achieves its design goal of mapping multimodal features into a unified space where they can be fused through addition without semantic shift, ensuring compatibility for downstream tasks under any modal failure. The implementation details and more results of t-SNE analysis can be founded in Section A.10.

## 4.4 QUALITATIVE ANALYSIS

Figure 5(a) shows that with enough LiDAR points, SiMO-PF has the same detection quality as Pyramid Fusion. When LiDAR becomes sparse or even fails, SiMO-PF (the upper row) can still achieve reasonable performance, as shown in (c). It also reveals the necessity of applying extra modality to prevent the dangers posed by LiDAR failure. The comparison in (d) indicates the lead of SiMO-PF (Camera), which

Table 4: Procrustes analysis.

| PROCRUSTES DISPARITY | BEVFUSION | BEFORE LAMMA | AFTER LAMMA |
|---|---|---|---|
| CAM VS LIDAR | 0.8645 | 0.6747 | **0.0472** |
| CAM VS FUSED | 0.7297 | 0.3886 | **0.0215** |
| LIDAR VS FUSED | 0.5747 | 0.2773 | **0.0064** |

is consistent with the above analysis that Pyramid Fusion does not fully utilize the extracted camera features. More visualizations can be found in Section A.5.

## 5 LIMITATIONS

While SiMO effectively addresses modality failure in collaborative perception, it exhibits certain limitations. First, single-modality operation performance is inherently limited by feature extractor capabilities. For instance, in single-view camera setups (e.g., DAIR-V2X), the lack of multi-view parallax limits depth estimation accuracy. SiMO preserves baseline performance but cannot overcome this physical information bottleneck. Second, our PAFR strategy, while ensuring deterministic convergence and mitigating modality competition, necessitates a multi-stage training pipeline. This inevitably increases the total training duration compared to end-to-end joint training methods. Finally, LAMMA's additive nature, designed to preserve feature independence, lacks the implicit smoothing inherent in convolution-based fusion. This potentially makes the system more sensitive to high-intensity sensor noise unless specific denoising modules are integrated.

## 6 CONCLUSION

This paper proposes SiMO to enable a multimodal collaborative perception system to function with various modal failures. SiMO believes that merging multimodal features in a unified feature space is key to solving the problem, and it specifically addresses the issue of modal competition to ensure that each modality branch retains its independent functionality. Through the plug-and-play LAMMA and PAFR training methods, SiMO effectively aligns multimodal features while preserving modality-specific features. This makes it easy to adapt SiMO to other multimodal approaches without requiring modifications to existing methods, giving SiMO broader versatility.

ACKNOWLEDGMENTS

This work was supported in part by the New Generation Artificial Intelligence-National Science and Technology Major Project 2025ZD0123703, in part by the National Natural Science Foundation of China under Grant 61936014 and 62076183, in part by the Chenguang Program of Shanghai Education Development Foundation and Shanghai Municipal Education Commission under Grant 21CGA24, in part by Shanghai Municipal Science and Technology Major Project No. 2021SHZDZX0100, in part by the Shanghai Science and Technology Innovation Action Plan Project 22511105300, in part by the National Natural Science Foundation of China under Grant U23A20382 and in part by Fundamental Research Funds for the Central Universities.

REPRODUCIBILITY STATEMENT

To ensure the transparency and reproducibility of this work, we have provided the complete codebase in the supplementary materials, which includes implementation details of our experiments, training

configurations, and data processing scripts. In addition, we have detailed the model architecture, hyperparameter settings, and all key steps in the training process in the appendix. We believe these materials will provide readers with sufficient information to easily reproduce our main results and conduct further research based on them.

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

# A APPENDIX

## A.1 DETAILED ARCHITECTURE SETTINGS

We present more architecture settings of SiMO-AF in Table 5 for readers to quickly obtain some detailed information about the implementation. The same details can be found in the `opencood/hypes_yaml/lidar_camera_lamma3_attfuse.yaml` file in the code. As for SiMO-PF, we adopt the exact settings of HEAL (Lu et al., 2024) to reuse their open-sourced model. The detailed settings can be found in `opencood/hypes_yaml/lidar_camera_lamma3_pyramid_fusion.yaml`.

Table 5: Detailed SiMO-AF settings.

| Modules | Submodules | Settings |
|---|---|---|
| PointPillar | Voxel Feature Encoder (VFE) | With normalization, absolute 3D coordinates, filters=64 |
| | PointPillar Scatter | Channels=64 |
| | BEV backbone | ResNet: layers=[3, 5, 8], strides=[2, 2, 2], filters=[64, 128, 256], upsample_strides=[1, 2, 4], upsample_filters=[128, 128, 128] |
| | Shrinker header | Convolutional layer: kernel_size=3, stride=1, channel shrunk from 384 to 256 |
| | Aligner | Convnext: blocks=3, channels=256 |
| LSS | Encoder | EfficientNet: img_feature=128, downsample=8, with depth supervision |
| | BEV backbone | ResNet: layers=[3, 5, 8], strides=[2, 2, 2], filters=[64, 128, 256], upsample_strides=[1, 2, 4], upsample_filters=[128, 128, 128] |
| | Shrinker header | Convolutional layer: kernel_size=3, stride=1, channel shrunk from 384 to 256 |
| | Aligner | Convnext: blocks=3, channels=256 |
| LAMMA | - | Feature_stride=2, feat_dim=256, dim=256 |
| AttFusion | - | Feat_dim=256 |
| Detect Head | - | Channels=256, anchors=2 |

## A.2 ALGORITHM DESCRIPTION OF LAMMA

Algorithm 1 presents the forwarding pipeline of our Length-Adaptive Multimodal Fusion module.

## A.3 TRAINING AND COMPUTATIONAL COST ANALYSIS

Our training strategy to mitigate modal competition inevitably extends training duration. The training duration required for each step is detailed in Table 6. It can be observed that the Align+Fusion+RD phase adds more training time than training a single branch independently. This is because, apart from the pre-trained feature extractors, all subsequent modules necessitate retraining and fine-tuning to preclude any modal bias.

Although this training strategy, based on a pre-trained branch, increases the overall training time, it offers greater certainty compared to grid-searching for the optimal training parameters of gradient modulation and forward dropout—and surprisingly, the total training time ends up being shorter. The longer training duration arises specifically when compared to methods that fail to address the issue of modal competition. We believe that extending the training period in a controlled manner to tackle the problem of modal competition represents the lowest-cost solution available.

Table 6: Training stages and epochs.

| STAGE | TRAINING EPOCHS |
|---|---|
| PRETRAIN (LIDAR_ATTFUSE) | 20 |
| PRETRAIN (CAM_ATTFUSE) | 20 |
| ALIGN (SIMO-AF (L)) | 15 |
| ALIGN (SIMO-AF (C)) | 5 |
| FUSION (SIMO-AF (L+C)) | 5 |
| RD FINE-TUNE | 5 |

Moreover, considering that the encoders are frozen in subsequent training, requiring only the training of downstream modules, this does not significantly increase the requisite VRAM. Conversely, due to the separate training of the encoder and fusion module, the VRAM requirements are, in fact, reduced, which enabled us to complete SiMO's training on an RTX 3090 GPU.

Model size and computational cost analysis is presented in Table 7. While LAMMA's parameters are considerably larger than BEVFusion's multimodal fusion module, they comprise only 6.02% of the overall model, less than a Shrink Conv Layer. Thus, we contend this does not unduly increase model size. Similarly, FLOPS comparison shows LAMMA's incorporation does not substantially elevate computational burden.

Table 7: Parameter count and FLOPS comparison.

| MODULE | SiMO-PF | BEVFUSION |
|---|---|---|
| LiDAR ENCODER (ENCODER_M1) | 768 | 768 |
| CAMERA ENCODER (ENCODER_M2) | 14,635,796 | 14,635,796 |
| LiDAR BACKBONE (BACKBONE_M1) | 226,176 | 226,176 |
| CAMERA BACKBONE (BACKBONE_M2) | 267,136 | 267,136 |
| ALIGNER (ALIGNER_M2) | 109,440 | 109,440 |
| MULTIMODAL FUSION (MM_FUSION) | 1,312,512 (LAMMA) | 73,856 (CONCAT+CONV) |
| PYRAMID BACKBONE (PYRAMID_BACKBONE) | 3,757,635 | 3,757,635 |
| SHRINK CONV (SHRINK_CONV) | 1,475,072 | 1,475,072 |
| DETECTION HEADS | 5,140 | 5,140 |
| TOTAL PARAMETERS | 21,789,675 | 20,551,019 |
| PROPORTION OF FUSION MODULE PARAMETERS | 0.0602 | 0.0036 |
| FLOPS | 223.95G | 219.31G |

## A.4 COMMUNICATION COST ANALYSIS

We adopted HEAL's Pyramid Fusion as our multi-agent fusion method without modification, consequently aligning our communication bandwidth and latency with HEAL's. For the convenience of the readers interested in this issue, we calculated SiMO's communication volume using the same methodology as Where2comm (refer to Appendix 7.5 in Hu et al. (2022)), with the following results:

$$\log_2\left(\frac{H \times W \times C \times 32}{8}\right) = \log_2\left(\frac{64 \times 64 \times 64 \times 32}{8}\right) = 20 \tag{5}$$

Here, H and W represent the height and width of the shared BEV features, respectively, and C is the number of feature channels. Multiplying by 32 indicates that features are stored in float32 format, and dividing by 8 yields the communication volume in Bytes. The final metric is the base-2 logarithm of this value. The shared BEV feature map in SiMO has a dimension of H=64, W=64, with C=64 channels.

For comparison, Where2comm's communication volume ranges from 13.84 to 25.93 (according to Table 4 in Hu et al. (2022)), demonstrating that SiMO's communication is also relatively concise. We believe that by integrating existing methods for reducing communication volume, SiMO's communication overhead can be further minimized in future work.

## A.5 MORE VISUALIZATIONS

From Figure 7, we can observe that by incorporating the camera modality, SiMO-AF's detection results have fewer false positives. AttFusion tends to misidentify objects similar in size to vehicles as targets, as shown in the enlarged areas of the detection results. However, by integrating image features, SiMO-AF can correctly ignore such interferences. Additionally, SiMO-AF is more likely to mistakenly judge that there are still targets behind the occluded area of the detected objects, leading to false positives. This situation often occurs in peripheral areas without the collaboration of other agents, indicating that collaborative perception can improve this condition.

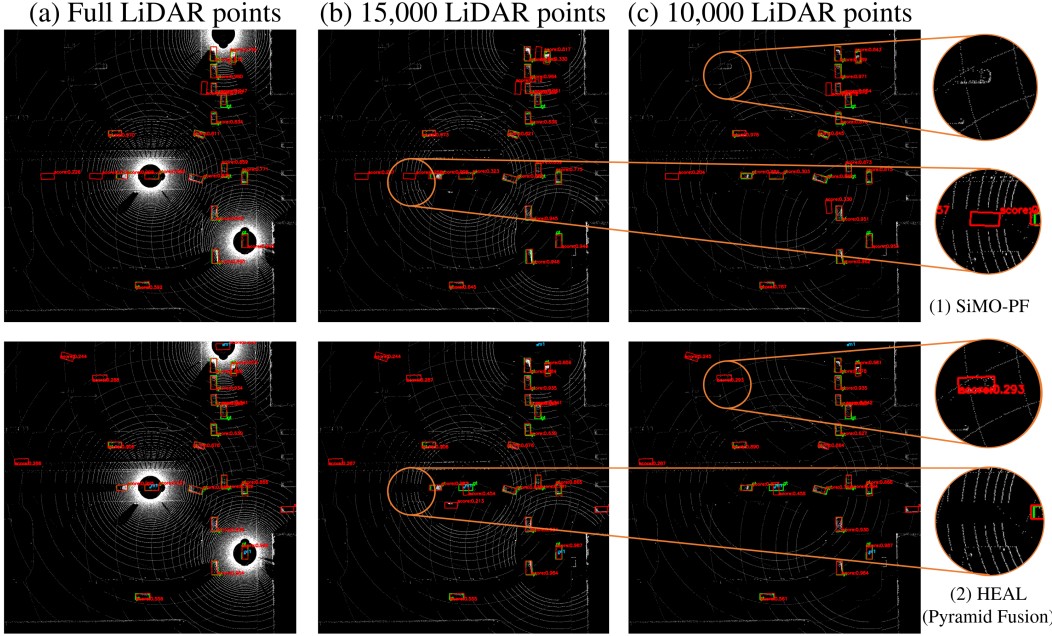

Figure 7: Visualization of SiMO-AF's (the 1st row) and AttFusion's (the 2nd row) detection results, ground truth and collaborative point clouds. (a)(b)(c) are detection visualizations with all, 15000 and 10000 LiDAR points. Typical detection errors are enlarged for analysis.

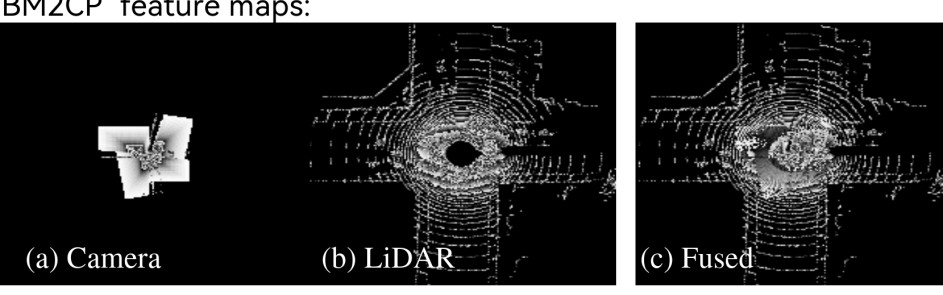

Figure 8: The visualizations of BM2CP's features. (a) and (b) are respectively the camera and LiDAR feature before fusion, and (c) is the fused feature combining two modals.

### A.6 THE LIMITATIONS OF BM2CP

From Table 1, it can be seen that the performance of BM2CP is somewhat inferior to that of AttFusion and Pyramid Fusion, even after combining two modal features. Here, we attempt to explore the reasons for this, but since the issue is outside the scope of this paper's research, we include the analysis in Appendix.

We infer that the insufficient performance of BM2CP is mainly due to two factors. Firstly, the original setup of BM2CP was not adequately optimized to select the best hyperparameters. In contrast, the AttFusion method used is the result of continuous optimization in subsequent work, including the encoder and decoder replacement but only left AttFuse module the same as the orignal, hence its performance is better than the original BM2CP. Additionally, we attempted to visualize the features of the two modalities before fusion and the fused features in BM2CP, as shown in Figure 8. We observe that compared to point cloud features, image features have significant locality. Their effective features are concentrated in the vicinity of the agent, so as the perception range expands, the gain

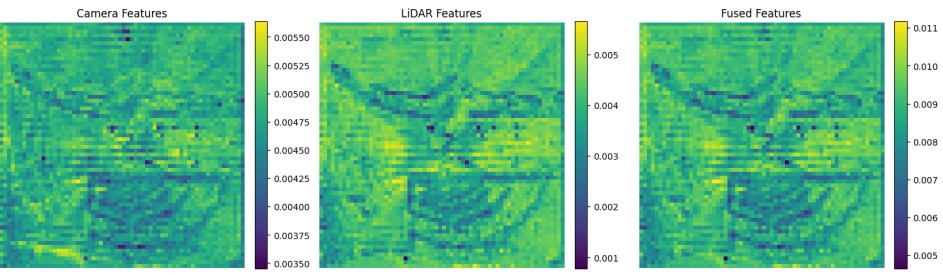

Figure 9: The visualizations of LiDAR and camera features aligned with LAMMA and the fused feature.

---

**Algorithm 1** LAMMA Forwarding

---

1: **Input**: Modality features $\mathbf{Z}_m \in \mathbb{R}^{c \times n}$ for $m \in \mathcal{M}$, where $\mathcal{M}$ is the set of modalities ($|\mathcal{M}| = M$)
2: **Parameter**: Positional encoding $\mathbf{p} \in \mathbb{R}^c$, Learnable matrices $\mathbf{W}_p \in \mathbb{R}^{d \times c}, \mathbf{W_Q} \in \mathbb{R}^{d \times d}, \mathbf{W_K} \in \mathbb{R}^{d \times d}, \mathbf{W_V} \in \mathbb{R}^{d \times d}, \mathbf{W}_{\text{out}} \in \mathbb{R}^{c \times d}$
3: **Output**: Fused feature $\mathbf{Z}_{mm} \in \mathbb{R}^{c \times n}$
4:
5: $\mathbf{Z}_{\text{concat}} \leftarrow [\,]$
6:   # Add Positional Encoding and Initial Projection
7: **for** $m \in \mathcal{M}$ **do**
8:     $\mathbf{Z}_m^{\text{pe}} \leftarrow \mathbf{Z}_m + \mathbf{p}$                          ▷ Add $c$-dimensional positional encoding to input features
9:     $\mathbf{Z}_m^{\text{proj}} \leftarrow \mathbf{W}_p \mathbf{Z}_m^{\text{pe}}$                                 ▷ Project features from $c$ to $d$
10:    $\mathbf{Z}_{\text{concat}} \leftarrow \left[ \mathbf{Z}_{\text{concat}}; \mathbf{Z}_m^{\text{proj}} \right]$      ▷ Concatenate projected features along the sample dimension $(d \times n \to d \times Mn)$
11: **end for**
12:
13:   # Cross-Modal Query Initialization
14: $\mathbf{Q} \leftarrow \mathbf{W_Q} \mathbf{Z}_{\text{concat}}$                                 ▷ Project to query space $\mathbf{Q} \in \mathbb{R}^{d \times Mn}$
15:
16:   # Multi-Head Attention Fusion Blocks
17: **for** $m \in \mathcal{M}$ **do**
18:     $\mathbf{K}_m \leftarrow \mathbf{W_K} \mathbf{Z}_m^{\text{proj}}$                          ▷ Project to key space $\mathbf{K}_m \in \mathbb{R}^{d \times n}$
19:     $\mathbf{V}_m \leftarrow \mathbf{W_V} \mathbf{Z}_m^{\text{proj}}$                          ▷ Project to value space $\mathbf{V}_m \in \mathbb{R}^{d \times n}$
20:     $\mathbf{Z}_{\text{att}}^{(m)} \leftarrow \text{MultiHead}\left(\mathbf{Q}, \mathbf{K}_m, \mathbf{V}_m\right)$    ▷ Apply Multi-Head Attention; output $d \times Mn$
21:     $\mathbf{Z}_{\text{att}}^{(m)} \leftarrow \text{LayerNorm}\left(\mathbf{Z}_{\text{att}}^{(m)} + \mathbf{Z}_{\text{concat}}\right)$    ▷ Add residual connection and LayerNorm
22:     $\mathbf{Z}_{\text{fused}}^{(m)} \leftarrow \text{LayerNorm}\left(\text{MLP}(\mathbf{Z}_{\text{att}}^{(m)}) + \mathbf{Z}_{\text{att}}^{(m)}\right)$    ▷ Apply MLP and LayerNorm
23:     $\mathbf{Z}_{\text{fused}}^{(m)} \leftarrow \text{Sum}(\text{Split}(\mathbf{Z}_{\text{fused}}^{(m)}))$    ▷ Split $d \times Mn$ into $M$ tensors of $d \times n$ and sum them
24: **end for**
25:
26:   # Aggregation and Final Projection
27: $\mathbf{Z}_{mm} \leftarrow \mathbf{W}_{\text{out}}\left(\sum_{m \in \mathcal{M}} \mathbf{Z}_{\text{fused}}^{(m)}\right)$    ▷ Sum modality features and project back to $c$ channels
28:
29: **return** $\mathbf{Z}_{mm}$                                     ▷ Return the fused feature

---

that image features bring to the fused features relatively decreases. This might be because BM2CP does not have a separate BEV backbone for image modality to encode and decode features, achieving feature alignment between the two modalities, but instead directly uses an attention mechanism to extract useful information for point cloud features from unaligned image features.

Therefore, we believe that using a separate BEV backbone to encode modality features into BEV features is beneficial for the semantic alignment of the two modality features and full utilization of the complementary information of multi-modalities. Moreover, aligning the semantics of multimodal features before fusion can bring a more effective fusion of modality information. For comparison, we have added visualizations of SiMO's features before and after fusion in Figure 9, which demonstrate that LAMMA effectively aligns the geometric structures of LiDAR and Camera features.

## A.7 VALIDATIONS ON MORE DATASETS

As an emerging field, collaborative perception still lacks a sufficient number of publicly available datasets. Among the existing datasets, only OPV2V (Xu et al., 2022c), V2XSet (Xu et al., 2022b), V2X-Sim (Li et al., 2022), DAIR-V2X (Yu et al., 2022) and V2XReal (Xiang et al., 2024) provide multimodal data, while open-sourced real-world datasets are limited to DAIR-V2X and V2VREAL. Here, we validate SiMO on V2XSet to extend it to the Vehicle-to-Everything (V2X) scenario, and further explore DAIR-V2X and V2XReal for real-world validation.

Table 8: 3D detection results on V2XSet of models trained with our learning strategy and RD (in %).

| Method | Modality | AP@30 | AP@50 | AP@70 |
|---|---|---|---|---|
| BEVFusion (Liu et al., 2023) | L+C (RD) | 83.56 | 82.30 | 65.45 |
| | L (RD) | 75.96 | 74.53 | 53.22 |
| | C (RD) | 0 | 0 | 0 |
| HEAL (Lu et al., 2024) (Pyramid Fusion) | L | 94.96 | 94.25 | 89.65 |
| | C | 56.99 | 46.72 | 31.71 |
| SiMO (AttFusion) | L+C (RD) | 86.95 | 85.20 | 72.09 |
| | L (RD) | 86.40 | 84.85 | 70.05 |
| | C (RD) | 52.10 | 39.81 | 19.39 |
| SiMO (Pyramid Fusion) | L+C (RD) | 93.46 | 92.66 | 85.85 |
| | L (RD) | 91.09 | 90.44 | 84.09 |
| | C (RD) | 68.16 | 56.42 | 33.89 |

**V2XSet**. V2XSet (Xu et al., 2022b) is a large-scale collaborative perception dataset that mirrors the data structure and simulation environment of OPV2V. While OPV2V focuses on collaborative vehicles, V2XSet expands the scope of collaboration to include road infrastructure. Similar to the vehicles, these infrastructures are equipped with four RGB cameras for frontal, rear, right, and left perspectives, as well as a LiDAR sensor.

Table 8 demonstrate that SiMO maintains its adaptability to modal failure despite the perspective divergence among collaborative agents in vehicle-infrastructure scenarios. Moreover, SiMO and BEVFusion behave similarly to the results on OPV2V. For instance, SiMO-PF perform better with camera only, even compared with HEAL in camera-only setting. and BEVFusion cannot adapt to modal failure with RD fine-tuning.

**DAIR-V2X**. DAIR-V2X (Yu et al., 2022) is the first large-scale real-world dataset designed for research on vehicle and road infrastructure collaboration. It includes the perception data of a vehicle and a road infrastructure, both equipped with a frontal RGB camera and a LiDAR sensor. Table 9 shows that after RD, SiMO-PF shows notable accuracy drops with LiDAR-only and dual modalities, while camera-only performance remains comparable to Pyramid Fusion. Similar results to other datasets suggest that SiMO may have adapted to modal failures, though the low accuracy of the camera-only approach makes this claim less reliable.

We attribute the poor camera-only results to DAIR-V2X's single-perspective setup. The LSS-based camera branch used in this study is designed for multi-view fusion, extracting valuable depth information from parallax. In the absence of parallax across multiple views, it becomes challenging to achieve convergence of the camera branch during **Step 1**. However, this issue does not stem from an inherent flaw in the SiMO fusion framework. Instead, SiMO's modular design allows for the

Table 9: 3D detection performances of SiMO on DAIR-V2X (in %).

| Method | Modality | AP@30 | AP@50 | AP@70 |
|---|---|---|---|---|
| HEAL (Lu et al., 2024) (Pyramid Fusion) | L | 77.06 | 69.35 | 42.74 |
| | C | 8.08 | 2.15 | 0.22 |
| SiMO (Pyramid Fusion) | L+C (w/o RD) | 74.56 | 64.51 | 29.31 |
| | L+C (RD) | 68.07 | 51.82 | 18.84 |
| | L (RD) | 66.89 | 52.33 | 19.35 |
| | C (RD) | 8.60 | 2.24 | 0.26 |

flexible integration of more advanced single-view 3D detectors in the future—a key direction for our ongoing research. Furthermore, to our best knowledge, existing camera-based methods for DAIR-V2X (e.g. ImVoxelNet (Yu et al., 2022), EMIFF (Wang et al., 2024b)) achieve a maximum AP@50 of 15.61%. This suggests that the inherent limitations of the camera data in this dataset are the root cause of the lower performance.

**V2XReal**. V2XReal (Xiang et al., 2024) is the latest real-world dataset containing a large scale of both multi-view RGB images and LiDAR point cloud in the V2X scenario. Although the dataset includes

both point cloud and image data along with annotations, the absence of examples for importing image data in their open-source code, coupled with the annotation errors in some examples, has prevented us from successfully validating our method on this dataset.

## A.8 THE EVALUATION OF ROBUSTNESS UNDER NOISY CONDITIONS

We introduced Gaussian noise with a mean of 0 and standard deviation $\sigma$ to the LiDAR point cloud coordinates to evaluate the robustness of SiMO under noisy conditions. The results in Table 10 reveals a fascinating and crucial design trade-off: in noise-free or low-noise ($\sigma$=0.1) environments, SiMO outperforms BEVFusion. However, as noise intensity escalates, SiMO's performance degrades more markedly than BEVFusion's.

Table 10: AP performance with LiDAR noise.

| METHOD | AP@30/50/70 | | | | |
|---|---|---|---|---|---|
| LIDAR NOISE | $\sigma = 0$ | $\sigma = 0.1$ | $\sigma = 0.5$ | $\sigma = 1$ | $\sigma = 2$ |
| BEVFUSION | 0.95/0.94/0.81 | 0.95/0.94/0.81 | 0.94/0.93/0.79 | 0.93/0.92/0.77 | 0.77/0.76/0.55 |
| SiMO-PF | 0.98/0.98/0.95 | 0.98/0.97/0.92 | 0.92/0.90/0.73 | 0.81/0.78/0.55 | 0.71/0.67/0.43 |

Our analysis is that BEVFusion's advantage stems from its convolutional fusion mechanism. Convolution, as a potent local spatial filter, inherently learns to smooth and suppress noise at the data point level, achieving an implicit denoising effect.

SiMO's characteristics are rooted in its core design philosophy. Our method, through 'attention + addition', is meticulously designed to ensure that modal features, once mapped to a shared space, largely retain their independence and integrity. While this design is paramount for addressing catastrophic modal failures (structural absence), it also implies that SiMO will 'trust' and faithfully propagate noise from individual modalities, as it lacks the local filtering/smoothing mechanisms inherent in convolution.

Regarding SiMO's performance degradation under high noise levels, this occurs because SiMO 'trusts' features from different modalities, overlooking potential discrepancies in their noise levels. This underscores the importance of clean features in multi-modal fusion. To integrate SiMO's robustness to macroscopic failures with specialized denoising modules to achieve more comprehensive perceptual resilience, we propose augmenting the encoder in each modal branch with convolutional layers, similar to those in BEVFusion, to smooth and filter noise. Concurrently, incorporating a suitable amount of random noise during training would enable the encoder to learn noise filtering, thereby extracting denoised features. This approach would allow the model to adapt to high-noise environments without requiring modifications to LAMMA itself.

## A.9 THE CONTRIBUTION OF EACH STEP IN OUR LEARNING STRATEGY

Our learning strategy to overcome modality competition includes 3 steps (**Step 4** is applied for adapting to modal failure), respectively corresponding to the training of feature extractors, aligners and common downstream modules (LAMMA, multi-agent module and task heads). We conduct ablation experiments to further explore the contributions of each step.

Since we inherited the pretrained AttFusion models as our branches, we start our training from **Step 2**, in which we focus on the training of Aligners $g_m$. Despite the well-trained feature extractors having

Table 11: The ablation study of our training method (measured in %).

| METHOD | MODE | AP@30 | AP@50 | AP@70 |
|---|---|---|---|---|
| ATTFUSION (XU ET AL., 2022C) | L | 96.68 | 95.09 | 87.16 |
| | C | 68.31 | 52.91 | 25.30 |
| SiMO-AF W/O ALIGNERS | L+C | 87.07 | 83.11 | 35.70 |
| SiMO-AF NAIVE TRAINING | L+C | 96.10 | 94.26 | 78.49 |
| SiMO-AF OUR TRAINING | L+C | 96.51 | 95.26 | 85.10 |

contributed to a good performance, the inherent semantic differences between modalities can confuse the multimodal fusion module and worsen the performance. Aligners $g_m$ are designed to map features from different modalities into feature spaces compatible with LAMMA, facilitating multimodal fusion. As shown in Table 11, SiMO-AF without aligners (in naively training) cannot converge to the same level as the best branch that is inherited from AttFusion's LiDAR model, indicating the necessity of aligners.

In **Step 3**, we freeze the branches to train the common modules. In comparison, a naive training is not involved in freezing any components. Table 11 shows that without freezing the branches to avoid modality competition, the performance still suffers from a decline of 8.76 in AP@70, even with the pretrained branches and aligners. Only SiMO-AF with aligners can deliver a comparable performance to the best branch with our learning strategy.

## A.10    T-SNE COMPARISONS

To ascertain whether multimodal features achieve alignment, we conducted a t-SNE analysis on the unimodal and fused features of BEVFusion and LAMMA. We preserved 100 sets of the following features: (1) Raw features from BEVFusion encoders and fused features; (2) Raw features from our encoders (prior to alignment) and fused features; (3) Aligned L features, Aligned C features, and fused features from LAMMA. Subsequently, to mitigate computational intensity, we initially reduced the dimensionality of these features to 100 dimensions using PCA, followed by t-SNE visualization, which yielded the results presented in Figure 10.

By comparing the t-SNE analysis results of BEVFusion and LAMMA, we can observe that both multimodal features from BEVFusion and LAMMA naturally form distinct clusters, highlighting a clear modality gap in their representations. However, after LAMMA alignment, the clusters of multimodal features and the fused feature cluster exhibit striking structural similarities—something that was not evident in the features of BEVFusion and before LAMMA alignment. We believe this phenomenon reflects effective modality alignment, a claim further supported by quantitative Procrustes analysis results.

Notably, this alignment approach contrasts sharply with traditional contrastive learning methods. In contrastive learning, the typical outcome is the merging of multimodal features into a single large cluster, with each modality adopting an identical distribution pattern. We attribute this result to contrastive learning's emphasis on minimizing inter-modality differences while emphasizing shared commonalities across modalities. Unfortunately, this process often leads to the loss of modality-specific features within the unified feature space—a critical drawback when it comes to enabling independent operation for individual modalities. For this reason, instead of relying on contrastive loss for modality alignment, we propose a novel training method that not only ensures effective alignment of multimodal features but also preserves the unique modality-specific features.

## A.11    THE EFFECT OF ALIGNER TRAINING ORDER

As mentioned in Section 3.3, SiMO is designed to treat two modalities equally without any preset bias. So in the **Step 2** of Section 3.4.2, the training order of aligners should not influence the adaptablity of modal failure. To confirm that, we train SiMO with two different orders–LiDAR first and camera first.

Table 12: The effect of aligners' training order (%).

| ORDER | SIMO-AF (LIDAR FIRST) | | | SIMO-AF (CAMERA FIRST) | | |
|---|---|---|---|---|---|---|
| MODE | L+C | L | C | L+C | L | C |
| AP@30 | 96.10 | 95.11 | 66.53 | 95.55 | 95.12 | 63.70 |
| AP@50 | 94.98 | 94.02 | 49.69 | 94.28 | 93.81 | 47.30 |
| AP@70 | 84.81 | 82.35 | 22.59 | 81.07 | 81.07 | 20.07 |

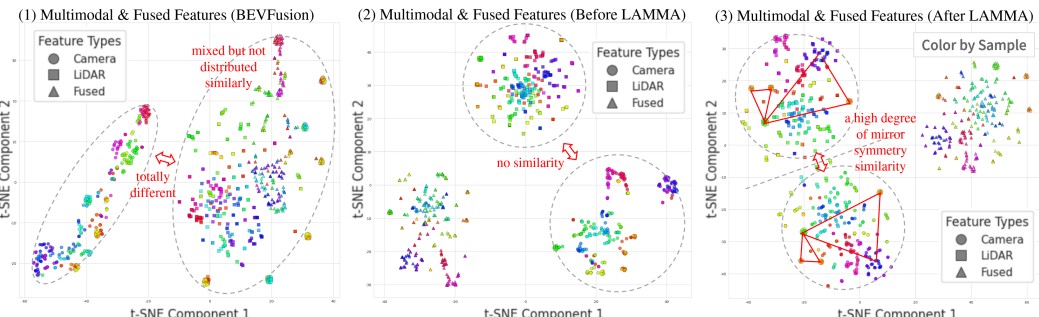

Figure 10: Only features after LAMMA (3) show a high degree of mirror symmetry similarity in t-SNE analysis. Features' distributions of BEVFusion (1) and before LAMMA (2) show no similarity.

Table 12 shows that SiMO can achieve the adaptability of modal failure regardeless of the training order of aligners.

Besides, we reversed the input order of two modalities in LAMMA, and the minimal performance changes (-0.30%/0.36%/1.32% for AP@30/50/70) further proves that LAMMA does treat two modal features consistently without any modality perference.

## A.12 THE EFFECT OF THE PROJECTION IN LAMMA

Since queries are concatenated for parallel computation in LAMMA, the memory usage surges in attention fusion, which makes LAMMA a memory bottleneck. The convolutional $\mathrm{Proj}$ and the transposed convolutional $\mathrm{Proj\_inv}$ in LAMMA aim to compress the BEV feature to process, saving the computation of Transformer block, and then restore the fused tokens back to BEV features. As the size of feature processed by Transformer is limited by GPU memory, $\mathrm{Proj}$ and $\mathrm{Proj\_inv}$ in fact determine the size of modal features $\mathbf{Z}_m$. Without the ability to flexibly adjust feature sizes using $\mathrm{Proj}$ and $\mathrm{Proj\_inv}$, a model with a fixed modal feature size of 64x64 cannot reach its maximum performance potential as shown in Table 13.

Table 13: The effect of feature size on detection precisions (in %).

| FEATURE SIZE | AP@30 | AP@50 | AP@70 |
|---|---|---|---|
| 128*128 | 96.51 | 95.26 | 85.10 |
| 64*64 | 95.48 | 93.78 | 78.50 |

Table 14: Performance comparison with varying agents in heterogeneous modalities.

| SETTING | HOMOGENEOUS MODALITY | | | HETEROGENEOUS MODALITY (L-EGO) | | | | | | HETEROGENEOUS MODALITY (C-EGO) | | | | | |
|---|---|---|---|---|---|---|---|---|---|---|---|---|---|---|---|
| MODALITY | L+C | L | C | ALL | CAV_0 (L) | +CAV_1 (C) | +CAV_2 (L) | +CAV_3 (C) | +CAV_4 (L) | ALL | CAV_0 (C) | +CAV_1 (L) | +CAV_2 (C) | +CAV_3 (L) | +CAV_4 (C) |
| AP@30 | 0.98 | 0.97 | 0.81 | 0.97 | - | 0.94 | 0.98 | 0.99 | 0.99 | 0.89 | - | 0.88 | 0.92 | 0.95 | 0.89 |
| AP@50 | 0.98 | 0.97 | 0.70 | 0.97 | - | 0.93 | 0.98 | 0.99 | 0.99 | 0.85 | - | 0.84 | 0.89 | 0.94 | 0.82 |
| AP@70 | 0.95 | 0.94 | 0.45 | 0.91 | - | 0.83 | 0.95 | 0.96 | 0.96 | 0.70 | - | 0.70 | 0.75 | 0.88 | 0.70 |

## A.13 THE EFFECT OF THE NUMBER OF CAVS

Following related works (Xiang et al., 2023; Lu et al., 2024), we conducted comparative experiments on varying CAV counts in heterogeneous failure scenarios. As shown in Table 14, SiMO's performance generally improves with more agents, especially in L-ego configurations, where collaborative performance nears its upper limit. Conversely, in C-ego settings, performance initially improves with more agents but declines at five CAVs. We posit this is due to the final agent being camera-only, whose features introduce greater uncertainty, interfering with multi-agent fusion. This suggests Pyramid Fusion could be enhanced by assessing agent uncertainty. However, as SiMO does not involve improvements to multi-agent fusion methods, we do not consider this issue a limitation of SiMO itself.

## A.14 THE USE OF LARGE LANGUAGE MODELS (LLMS)

The study utilized LLMs to refine and enhance the paper based on the author's original manuscript, as well as to make partial code modifications and assist in bug fixes.

