# OpenReview forum: "SiMO: Single-Modality-Operable Multimodal Collaborative Perception"
_ICLR.cc/2026/Conference — ICLR 2026 Poster_

### Official Review · Reviewer_nBTy · 2025-10-27

**Soundness:** 3
**Presentation:** 3
**Contribution:** 3
**Rating:** 6
**Confidence:** 4

**Summary:**

This paper introduces SiMO (Single-Modality-Operable Multimodal Collaborative Perception), a novel framework designed to enhance the robustness and reliability of multi-agent collaborative perception (MACP) systems against sensor failures. The central problem SiMO addresses is that existing multimodal fusion methods in MACP are fragile, similar to a "series circuit,” failing completely when a critical sensor like LiDAR is unavailable.SiMO's solution functions like a "parallel circuit," allowing the system to operate with any single, effective modality. It achieves this through two main innovations: 1) Length-Adaptive Multi-Modal Fusion (LAMMA): This plug-and-play fusion module first aligns features from different modalities (LiDAR and Camera) and then integrates them using additive fusion to maintain a consistent feature space before and after fusion. 2) Pretrain-Align-Fuse-RD (PAFR) Training Strategy: This strategy addresses modality competition to ensure that each individual sensor branch is comprehensively and independently trained, thus strengthening the performance of the single-modality operations.

**Strengths:**

1. High Robustness Against Modal Failures: SiMO explicitly address the complex challenge of dynamic, heterogeneous modal failures in Multi-Agent Collaborative Perception, which is important for real-world safety-critical deployment of V2V.
2. The core idea of aligning and fusing features using addition to ensure semantic consistency between pre- and post-fusion feature spaces is straight-forward and effective.
3. Proposed LENGTH-ADAPTIVE MULTIMODAL FUSION (LAMMA) can handle varying input modalities. With RD training, the performance of camera-only collaboration gets better, indicating an alleviation in modality competition.

**Weaknesses:**

1. This framework necessitates multiple stage training, which increases the training complexity.
2. Late fusion metrics are not reported.
3. The results on DAIR-V2X is not persuasive. While authors attribute the poor performance to the single-camera sensor and out-of-date camera-based detector, it would be encouraging to implement a more modern camera detector to validate the SiMO's effectiveness.

**Questions:**

no questions.

---

> ### Author Response · Authors · 2025-11-20
>
> We are profoundly grateful for your support of our endeavors. We hope that the supplementary experiments and explanations provided below will address your inquiries, we would be delighted to elaborate further should you have any more questions.
>
> **Weaknesses:**
>
> > 1. This framework necessitates multiple stage training, which increases the training complexity.
>
> We appreciate your concern regarding training complexity. We acknowledge that SiMO involves a multi-stage pipeline; however, this is a deliberate design trade-off to ensure **training determinism** and **modality independence**.
>
> While existing joint-training methods theoretically offer a "one-stage" solution, they often suffer from modality competition. Addressing this typically requires extensive hyperparameter tuning (e.g., balancing loss weights or gradient modulation) via grid search, which is computationally expensive and lacks guaranteed convergence.
>
> In contrast, our "Pretrain-Align-Fuse-RD" strategy decouples feature extraction from fusion. This converts the uncertain cost of hyperparameter tuning into a **deterministic and controllable computational cost**. Furthermore, once the extractors are trained, they are frozen, making the subsequent training of the fusion module lightweight. We believe this "divide-and-conquer" approach is a more robust and practically efficient solution for the complex task of collaborative perception.
>
>
> > 2. Late fusion metrics are not reported.
>
> We thank the reviewer for suggesting this comparison. To isolate the impact of the collaboration scheme, we have added results for Late Fusion and a No-Collaboration baseline:
>
> | Collaboration Method                 | AP@30 | AP@50 | AP@70 |
> | -------------------------------------- | ------- | ------- | ------- |
> | No Collaboration (SiMO w/o Pyramid Fusion)    | 96.95 | 96.16 | 87.88 |
> | Late Fusion (SiMO w/o Pyramid Fusion)    | 98.65 | 98.31 | 94.76 |
> | Intermediate Fusion (SiMO) | 98.38 | 98.05 | 94.89 |
>
> As shown, SiMO achieves competitive performance comparable to Late Fusion. The relatively small gap between fusion strategies is partly due to the saturation of single-agent performance on this specific dataset. However, the core advantage of SiMO lies in **robustness** rather than just peak performance. Unlike Late Fusion, which depends on the quality of object proposals (and fails if a modality misses the object entirely), SiMO fuses information at the feature level. This allows the system to recover objects that might be missed by single-modality proposals due to occlusion or sensor noise, providing a more resilient system structure.
>
> > 3. The results on DAIR-V2X is not persuasive. While authors attribute the poor performance to the single-camera sensor and out-of-date camera-based detector, it would be encouraging to implement a more modern camera detector to validate the SiMO's effectiveness.
>
> We appreciate this insightful comment. We acknowledge the lower performance on DAIR-V2X but attribute it primarily to the inherent **physical limitations of the dataset**, specifically the lack of multi-view parallax which is critical for depth estimation in camera-based 3D detection.
>
> This is a dataset-wide challenge, not a flaw in SiMO. As referenced, even SOTA camera-based methods on DAIR-V2X, such as EMIFF (2024 ICRA) [1], achieve only 15.61% AP@50, while advanced detectors like BEVDepth and BEVFormer yield \<10% (reported by EMIFF).
>
> To validate SiMO's effectiveness, the key metric is whether it **preserves the performance of the individual branch** within a fusion framework. Our results show that SiMO (Camera-only) matches and slightly exceeds the performance of the baseline HEAL (Camera-only) (8.60% vs. 8.08% AP@30). This aligns with our observations across both the OPV2V-H and V2X-Set datasets. The inability to adapt to modality failure should, as demonstrated by BEVFusion in OPV2V-H, result in complete failure (AP=0) in camera-only scenarios. Therefore, it confirms that our alignment strategy successfully enables the weaker modality to function independently, without being suppressed by the dominant LiDAR branch.
>
> Furthermore, SiMO is designed to be model-agnostic. While implementing a new SOTA monocular detector is beyond the scope of this work (which focuses on the *fusion mechanism*), SiMO can seamlessly integrate with stronger backbones as they become available. We have expanded the limitation section to clarify this sensor constraint.
>
> **Reference**
> [1] Wang, Z., Fan, S., Huo, X., Xu, T., Wang, Y., Liu, J., ... & Zhang, Y. Q. (2024, May). Emiff: Enhanced multi-scale image feature fusion for vehicle-infrastructure cooperative 3d object detection. In 2024 IEEE International Conference on Robotics and Automation (ICRA) (pp. 16388-16394). IEEE.

---

### Official Review · Reviewer_gwXL · 2025-10-28

**Soundness:** 3
**Presentation:** 4
**Contribution:** 3
**Rating:** 8
**Confidence:** 5

**Summary:**

This work presents the single-modality-operable multimodal collaborative perception framework, which is the first attempt to address modality failure in the context of multi-agent collaborative perception. The proposed SiMO employs Length-Adaptive Multi-Modal Fusion to effectively leverage the remaining modal features during modality failures while preserving semantic consistency across modalities. To reduce modality bias in modality competition, SiMO further proposes a “Pretrain–Align–Fuse–RD” training paradigm. The paper provides thorough analyses and extensive experiments, validating the effectiveness of the proposed approach.

**Strengths:**

1. This work is the first to address the problem of modality failure in the context of multi-agent collaborative perception, and it proposes an effective and practical solution through the SiMO framework.
2. The paper is supported by extensive experiments, complemented by insightful visualizations and in-depth data analyses.
3. The manuscript is clearly written and logically structured and the technical content is easy to follow.

**Weaknesses:**

1. The paper does not clearly explain the unique challenges of modality failure in the multi-agent collaborative perception (MACP) setting compared to the single-agent case. Similar modality failures could also appear in single-agent multi-modality scenarios. The authors should clearly explain why existing single-agent modality-robust methods are inadequate in the MACP context, and ideally include comparative experiments to substantiate this claim.

2. The training procedure is complex. It requires four separate end-to-end training stages, resulting in a large training overhead. The authors should provide a detailed analysis of its training robustness and efficiency.

3. The description of “heterogeneous model failure” in Table 2 is unclear, making it difficult to understand what modalities the ego vehicle and neighboring agents respectively possess in this setting.

**Questions:**

1. In the training process, Step 2 is sequential, aligning LiDAR first and then camera. This actually favors camera performance. My question is : when more modalities need to be aligned, how would this ordering be determined?

2. What is the exact input and what are the parameters used for the t-SNE visualization? Please clarify the implementation details.

---

> ### Author Response · Authors · 2025-11-20
>
> We extend our sincere gratitude for your invaluable support and insightful suggestions, which have greatly contributed to the refinement of this article. Below, we address each of your questions individually.
>
> **Weaknesses:**
>
> > 1. The paper does not clearly explain the unique challenges of modality failure in the multi-agent collaborative perception (MACP) setting compared to the single-agent case. Similar modality failures could also appear in single-agent multi-modality scenarios. The authors should clearly explain why existing single-agent modality-robust methods are inadequate in the MACP context, and ideally include comparative experiments to substantiate this claim.
>
> We appreciate this insightful question. While modality failure occurs in both settings, the **Multi-Agent Collaborative Perception (MACP)**  context introduces a dimension of complexity that existing single-agent methods cannot address.
>
> In a single-agent setting, robustness methods (like UniBEV or MetaBEV) only need to ensure that the fused features—or single modality features—are compatible with the local task head (**Intra-agent Alignment**). However, in MACP, features must be transmitted and fused across different agents. This introduces the unique challenge of **Heterogeneous Modal Failure**—a scenario where, for example, the Ego vehicle relies on LiDAR while a connected Neighbor relies on a Camera. This requires a stricter **Inter-agent Alignment**: *all* modality features from *all* agents must be mapped to a strictly unified semantic space. If the features are not perfectly aligned, the multi-agent fusion module will fail to integrate the heterogeneous information. Existing single-agent methods, which often rely on simple modality dropout (RD) without explicit feature space unification, fail to guarantee this interoperability.
>
> To substantiate this claim, we implemented UniBEV within our MACP framework. We used its proposed CNW (Channel-Normalized Weights) fusion and trained it with Modality Dropout (RD), mirroring the setup used for SiMO.
> |UniBEV (CNW fusion) w RD|AP@30|AP@50|AP@70|
> | --------------------------| -------| -------| -------|
> |L+C|93.33|91.71|70.75|
> |L|93.32|91.73|70.78|
> |C|1.93|0.0|0.0|
>
> Similar to standard BEVFusion, UniBEV suffers from a complete collapse of the Camera branch (AP@50 \$\approx\$ 0.0) in the MACP setting, despite being trained with RD.
>
> - **Why did it fail?**  Without our **PAFR (Pretrain-Align-Fuse-RD)**  strategy to strictly isolate and align branches, the Modality Competition in the complex MACP pipeline causes the model to over-rely on LiDAR features, suppressing the Camera branch.
> - **Why SiMO works?**  SiMO ensures that Camera features are semantically equivalent to LiDAR features *before* they enter the multi-agent fusion module. This allows SiMO to handle **Heterogeneous Modal Failure** (as detailed in Sec 4.2) seamlessly, a capability that single-agent methods transferred to MACP simply do not possess.
>
> In conclusion, existing single-agent methods are inadequate for MACP because they cannot prevent branch collapse in the face of complex multi-agent feature interaction. SiMO's combination of **LAMMA** (structural fusion) and **PAFR** (alignment training) is essential to solve this problem.

---

> ### Author Response · Authors · 2025-11-20
>
> **Weaknesses:**
> > 2. The training procedure is complex. It requires four separate end-to-end training stages, resulting in a large training overhead. The authors should provide a detailed analysis of its training robustness and efficiency.
>
> We acknowledge that SiMO involves a multi-stage training pipeline; however, this is a deliberate design choice to prioritize **determinism and controllability** over the uncertainty of traditional joint training.
>
> - **Hidden vs. Explicit Costs:**  Existing methods address modality competition by balancing learning rates or gradients. While theoretically elegant, they often require extensive **grid search** to find the optimal balance, creating a massive "hidden" temporal cost and instability. Our "Pretrain-Align-Fuse-RD" strategy converts this uncertainty into a deterministic computational cost (approx. 2-3x duration). We believe this predictable overhead is a more economic solution for the notoriously difficult task of collaborative perception.
>
> - **Computational Efficiency:**  It is worth noting that since we freeze converged modules sequentially, the **peak VRAM requirement** is actually lower than training a full heavy model end-to-end.
>
> - **Robustness:**  As shown in our ablation studies, this strategy guarantees that each modal branch reaches its performance ceiling independently, ensuring the robustness of the final system—a benefit that outweighs the training time overhead.
>
>
>
> > 3. The description of “heterogeneous model failure” in Table 2 is unclear, making it difficult to understand what modalities the ego vehicle and neighboring agents respectively possess in this setting.
>
> We apologize for the ambiguity. We have clarified the definition in the revised paper.
>
> "Heterogeneous modal failure" refers to a scenario where different agents in the collaboration graph lose different sensors. In our experiments, we simulate this using an **alternating failure pattern**:
>
> - **Agent 1 (Ego):**  Modality A
> - **Agent 2:**  Modality B
> - **Agent 3:**  Modality A
> - ... and so on.
>
> Since the collaborative performance is most sensitive to the Ego vehicle's status, we report results based on the Ego's active modality. For example, "L-Ego" implies the Ego vehicle uses LiDAR (while neighbors alternate), and "C-Ego" implies the Ego uses Camera.
>
> **Questions:**
>
> > 1. In the training process, Step 2 is sequential, aligning LiDAR first and then camera. This actually favors camera performance. My question is : when more modalities need to be aligned, how would this ordering be determined?
>
> This is an insightful question. We investigated the impact of alignment order in **Appendix A.11**.
>
> 1. **Experimental Observation:**  Our results (Table 12) show that while the performance gap is small, the "LiDAR-First" strategy yields slightly better results than "Camera-First".
> 2. **The "Anchor" Hypothesis:**  We argue that this is not because the sequential process favors the *second* modality, but rather because the *first* modality acts as the **Semantic Anchor**. LiDAR, with its precise 3D geometric structure, serves as a superior anchor. Aligning the Camera features to this high-quality LiDAR 3D space (LiDAR-First) is more effective than trying to align LiDAR features to a semantically ambiguous Camera space (Camera-First).
> 3. **Scalability Strategy (Anchor + Parallel):**  Based on this insight, for \$N \> 2\$ modalities (e.g., adding Radar), we propose an  **"Anchor + Parallel"**  strategy rather than a strictly sequential one:
>
>     - **Step 1:**  Select the modality with the best spatial representation (e.g., LiDAR) as the fixed Anchor.
>     - **Step 2:**  Train Aligners for all other modalities (Camera, Radar, Thermal) **in parallel**, mapping each independently to the Anchor space.
>
> This strategy avoids the complexity of determining a sequential order and prevents error accumulation, ensuring scalability.
>
>
> > 2. What is the exact input and what are the parameters used for the t-SNE visualization? Please clarify the implementation details.
>
> We have added the implementation details in the Appendix.
>
> - **Input Data:**  The t-SNE visualization inputs are the feature vectors taken from the BEV feature maps. specifically:
>
>   - **Figure 6:**  Inputs are the **Aligned** L features, **Aligned** C features, and  Fused features from LAMMA.
>   - **Figure 9:**  Compares three stages: (1) Raw features from BEVFusion encoders and fused feature ; (2) Raw features from our encoders (before alignment) and fused feature; (3) As Figure 6.
> - **Preprocessing:**   We applied **PCA** to reduce the feature dimensionality to 100 before feeding them into t-SNE to reduce computational cost. (Note: Raw features in Procrustes analysis were not PCA-reduced).
>
> We hope that the supplementary experiments and explanations provided below will address your inquiries, we would be delighted to elaborate further should you have any more questions.

---

### Official Review · Reviewer_AEwg · 2025-10-31

**Soundness:** 3
**Presentation:** 3
**Contribution:** 2
**Rating:** 4
**Confidence:** 4

**Summary:**

The paper proposes SiMO (Single-Modality-Operable Multimodal Collaborative Perception), a multimodal collaborative-perception framework that (1) aligns modality BEV features into a unified semantic space, (2) fuses them with a Length-Adaptive Multi-Modal Fusion (LAMMA) module that is intended to gracefully degrade to self-attention when a modality is missing, and (3) applies a Pretrain–Align–Fuse–RD (PAFR) training schedule to avoid modality competition (pretraining branches, training aligners, freeze branches, then fuse + random-drop fine-tuning). Experiments on OPV2V-H and V2XSet (plus DAIR-V2X discussion) show that SiMO maintains reasonable detection performance with LiDAR failure / camera-only operation while prior multimodal fusion baselines collapse.

**Strengths:**

The paper is, to the best of my knowledge, the first work in collaborative perception that explicitly tackles multimodal perception failure caused by missing modalities, with a particular focus on ensuring operability when only RGB inputs are available.

It identifies the inconsistency between pre-fusion and post-fusion features as the main reason for performance collapse during modality failure, and introduces LAMMA, a length-adaptive fusion module designed to maintain semantic consistency under varying modality combinations.

Through extensive experiments on multiple V2X collaborative datasets, the paper shows that each single-modality branch can function independently while maintaining competitive multi-modal performance, indicating that SiMO achieves strong robustness and communication efficiency under degraded sensory conditions.

**Weaknesses:**

The real-world validation is limited, and the camera results are inconsistent. On DAIR-V2X, the camera-only performance remains poor (the authors attribute this to the limitations of single-view LSS). This weakens SiMO’s claim of addressing single-modality operability in practice. The paper should (a) include stronger single-view camera baselines, or (b) tone down the claims regarding the single-view camera setting.

Comparisons with stronger or more recent modality-failure methods adapted to multi-agent settings (e.g., MetaBEV/UniBEV, contrastive alignment approaches, or simply stronger camera-only backbones) are recommended.

The scalability to more than two modalities and a large number of agents is unclear. How does the method perform with three or more modalities? Since the complexity varies with the number of connected queries, this should be discussed and, if possible, demonstrated experimentally.

Pose errors and asynchronous agents are not studied, yet collaborative perception is known to be sensitive to pose noise and temporal misalignment. The robustness to pose calibration errors or time delays should be evaluated or at least discussed (as prior works such as HEAL have shown this sensitivity).

The evaluation conducted solely on simulated datasets is not sufficiently convincing.

**Questions:**

In Figure 8, BM2CP’s feature visualizations are shown — why not visualize the features of your own method?

---

> ### Author Response · Authors · 2025-11-20
>
> We are deeply appreciative of your meticulous observations and insightful recommendations regarding our methodology.
>
> **Weaknesses:**
>
> > 1. The real-world validation is limited, and the camera results are inconsistent. On DAIR-V2X, the camera-only performance remains poor (the authors attribute this to the limitations of single-view LSS). This weakens SiMO’s claim of addressing single-modality operability in practice. The paper should (a) include stronger single-view camera baselines, or (b) tone down the claims regarding the single-view camera setting.
>
> We appreciate the reviewer's rigorous assessment. Regarding the camera performance on DAIR-V2X, we would like to clarify that the limitation stems primarily from the **physical constraints of the dataset** rather than the SiMO architecture. DAIR-V2X provides only single-view images, lacking the multi-view parallax essential for accurate depth estimation in BEV-based methods.
>
> This is a known bottleneck: even SOTA methods on DAIR-V2X, such as EMIFF [1], achieve only 15.61% AP@50, while advanced detectors like BEVDepth and BEVFormer achieve \<10%. This indicates that simply swapping the encoder does not overcome the lack of geometric information.
>
> The critical metric for SiMO's validity is **performance preservation under modality failure**. As shown in our results, SiMO (Camera-only) achieves parity with, and even slightly outperforms, the HEAL baseline (8.60% vs. 8.08% AP@30). This aligns with our observations across both the OPV2V-H and V2X-Set datasets. The inability to adapt to modality failure should, as demonstrated by BEVFusion in OPV2V-H, result in complete failure (AP=0) in camera-only scenarios. So this confirms that SiMO successfully aligns and utilizes the available features—however weak they may be—without structural collapse.
>
> We agree with your suggestion (b). We have revised the paper to explicitly state that while SiMO ensures operability, the absolute performance in single-view scenarios is bounded by the inherent limitations of monocular depth estimation.
>
>
> > 2. Comparisons with stronger or more recent modality-failure methods adapted to multi-agent settings (e.g., MetaBEV/UniBEV, contrastive alignment approaches, or simply stronger camera-only backbones) are recommended.
>
> We appreciate this constructive suggestion. We integrated UniBEV's core **CNW (Channel-Normalized Weights)**  fusion module and trained the model using its original **Modality Dropout (RD)**  strategy. The results are as follows:
> |UniBEV (CNW fusion) w RD|AP@30|AP@50|AP@70|
> | --------------------------| -------| -------| -------|
> |L+C|93.33|91.71|70.75|
> |L|93.32|91.73|70.78|
> |C|1.93|0.0|0.0|
>
> The results show that UniBEV suffers from a **complete collapse** of the Camera branch (AP \$\\approx\$ 0), similar to the standard BEVFusion baseline. This demonstrates that simply applying robust fusion architectures (like CNW) or Modality Dropout strategy is insufficient to handle the intense **modality competition** present in multi-agent collaborative perception. Without explicit alignment constraints, the model over-relies on the dominant LiDAR modality.
>
> For more analysis regarding the unique challenges of MACP compared to single-agent settings, please refer to our response to [Reviewer gwXL W1](https://openreview.net/forum?id=h0iRgjTmVs&noteId=JRXEiFHZp8).
>
>
> > 3. The scalability to more than two modalities and a large number of agents is unclear. How does the method perform with three or more modalities? Since the complexity varies with the number of connected queries, this should be discussed and, if possible, demonstrated experimentally.
>
> > 5. The evaluation conducted solely on simulated datasets is not sufficiently convincing.
>
> We acknowledge that our evaluation focuses on LiDAR and Camera due to the current scarcity of multi-modal collaborative perception datasets. Most existing benchmarks (OPV2V, V2XSet) are limited to these two modalities. While new datasets like V2X-Radar [2] are emerging, they are not yet fully accessible for benchmarking.
>
> However, **SiMO is scalable by design**. LAMMA treats modality features as a sequence of equal blocks within an attention mechanism. Theoretically, adding a third modality (e.g., Radar) involves simply projecting it into the common feature space and concatenating it to the query sequence, without requiring structural changes to the fusion block.
>
> Regarding real-world validation, we utilized DAIR-V2X, which is currently the only viable option (as V2X-Real [3] suffers from annotation issues). Despite its limitations, our experiments on DAIR-V2X demonstrate our commitment to validating SiMO in real-world settings. We are eager to extend SiMO to more modalities as the community's data infrastructure matures.

---

> ### Author Response · Authors · 2025-11-20
>
> **Weaknesses:**
> > 4. Pose errors and asynchronous agents are not studied, yet collaborative perception is known to be sensitive to pose noise and temporal misalignment. The robustness to pose calibration errors or time delays should be evaluated or at least discussed (as prior works such as HEAL have shown this sensitivity).
>
> Thank you for raising the issue of robustness. It is important to note that pose errors and synchronization delays primarily affect the **Multi-Agent Fusion** stage, rather than the Multimodal Fusion stage targeted by SiMO. Since SiMO integrates the established Pyramid Fusion (from HEAL) for agent collaboration, it inherits the robustness characteristics of that framework.
>
> To verify this empirically, we evaluated SiMO under varying levels of pose noise (Gaussian noise with std deviation \$\\sigma\$):
>
> | σ_std | AP@30 | AP@50 | AP@70 |
> | -------- | ------- | ------- | ------- |
> | 0      | 98.38 | 98.04 | 94.90 |
> | 0.2    | 98.22 | 97.12 | 77.33 |
> | 0.4    | 95.55 | 85.23 | 47.06 |
> | 0.6    | 86.55 | 71.47 | 35.76 |
>
> The results show that SiMO maintains high performance at reasonable noise levels (e.g., \$\\sigma\=0.2\$) and degrades as error increases, following a pattern similar to the original HEAL. This confirms that integrating LAMMA does not obviously compromise the system's tolerance to pose misalignment.
>
>
> **Questions:**
>
> > In Figure 8, BM2CP’s feature visualizations are shown — why not visualize the features of your own method?
>
> The visualization of BM2CP in Figure 8 was intended as a diagnostic tool to explain its suboptimal performance, specifically highlighting the semantic misalignment between its unaligned modalities. This observation motivated our design of the alignment module in SiMO.
>
> Addressing your suggestion, we have added visualizations of SiMO’s features before and after fusion in the Appendix of the revised paper. These visualizations demonstrate that LAMMA effectively aligns the geometric structures of LiDAR and Camera features. This qualitative result is further corroborated by our quantitative Procrustes analysis (Table 4), which shows a dramatic reduction in feature disparity (from 0.6747 to 0.0472) after alignment.
>
> We hope that the supplementary experiments and explanations provided below will address your inquiries, we would be delighted to elaborate further should you have any more questions.
>
>
> **Reference**
> [1] Wang, Z., Fan, S., Huo, X., Xu, T., Wang, Y., Liu, J., ... & Zhang, Y. Q. (2024, May). Emiff: Enhanced multi-scale image feature fusion for vehicle-infrastructure cooperative 3d object detection. In 2024 IEEE International Conference on Robotics and Automation (ICRA) (pp. 16388-16394). IEEE.
> [2] Yang, L., Zhang, X., Li, J., Wang, C., Ma, J., Song, Z., ... & Lv, C. (2024). V2x-radar: A multi-modal dataset with 4d radar for cooperative perception. arXiv preprint arXiv:2411.10962.
> [3] Xiang, H., Zheng, Z., Xia, X., Xu, R., Gao, L., Zhou, Z., ... & Ma, J. (2024, September). V2x-real: a largs-scale dataset for vehicle-to-everything cooperative perception. In European Conference on Computer Vision (pp. 455-470). Cham: Springer Nature Switzerland.

---

> > ### Comment · Reviewer_AEwg · 2025-11-25
> >
> > Thank you for your response. The clarifications and the comprehensive experiments have addressed most of my concerns. I will raise my score

---

> > > ### Author Response · Authors · 2025-11-25
> > >
> > > We sincerely thank you for your constructive comments and for recognizing our efforts in the rebuttal. Your insights were invaluable in strengthening the paper, and we are grateful for your support in raising the score.

---

### Official Review · Reviewer_Ntee · 2025-11-04

**Soundness:** 3
**Presentation:** 3
**Contribution:** 3
**Rating:** 6
**Confidence:** 3

**Summary:**

This paper proposes SiMO to address a critical yet underexplored challenge in multimodal collaborative perception: system failure under partial modality loss. SiMO introduces two key components: i) Length-Adaptive Multi-Modal Fusion: a fusion module that handles variable numbers of input modalities via attention mechanisms and maintains semantic consistency before and after fusion through additive combination. ii) Pretrain-Align-Fuse-RD training strategy: a staged training protocol that mitigates modality competition by pretraining and aligning modality-specific branches independently before joint fusion, followed by fine-tuning with random modality dropout (RD). Experiments show that SiMO achieves state-of-the-art performance in full multimodal settings while maintaining strong single-modality performance.

**Strengths:**

-	The paper tackles robustness to sensor failure in collaborative perception, which is a realistic and safety-critical issue that is rarely explored in existing multimodal collaborative perception literature.
-	The method is plug-and-play and demonstrated on two backbone frameworks (AttFusion and Pyramid Fusion), suggesting broad applicability.
-	Strong empirical validation: the paper presents comprehensive experiments across homogeneous/heterogeneous modality failures.

**Weaknesses:**

-	While RD improves robustness as a data augmentation strategy, the 0.5 dropout probability is not justified theoretically or empirically. A sensitivity analysis would strengthen this design choice.
-	While LAMMA fusion is designed to handle missing modalities, how does it differ from simpler alternatives—such as replacing the missing modality’s feature with a zero tensor in existing fusion schemes?

**Questions:**

See weakness.

---

> ### Author Response · Authors · 2025-11-20
>
> We wish to begin by expressing our appreciation for your positive assessment of our endeavors.
>
> **Weaknesses:**
>
> > 1. While RD improves robustness as a data augmentation strategy, the 0.5 dropout probability is not justified theoretically or empirically. A sensitivity analysis would strengthen this design choice.
>
> We appreciate the reviewer's scrutiny regarding the hyperparameter choice. To rigorously justify the 0.5 probability, we conducted a two-fold sensitivity analysis covering both **Modality Balance (LiDAR Dropout Ratio)**  and **Training Intensity (Overall Dropout Ratio)** .
>
> **1. Why LiDAR Dropout Ratio**  **=**  **0.5? (Balancing Between Modalities)**
>
> Assuming the overall probability of dropping a modality is fixed, we varied the conditional probability of dropping LiDAR vs. Camera.
>
> | LiDAR Dropout Ratio | L+C AP@30/50/70   | L AP@30/50/70     | C AP@30/50/70     |
> | --------------------- | ------------------- | ------------------- | ------------------- |
> | 0.3                 | 98.37/97.98/94.87 | 97.08/96.81/94.22 | 72.76/66.87/46.83 |
> | 0.5                 | 98.30/97.94/94.64 | 97.32/97.07/94.06 | 80.81/69.63/44.82 |
> | 0.7                 | 98.43/98.02/94.73 | 97.28/97.02/94.44 | 77.85/70.16/49.43 |
>
> - **Maximum Entropy Principle:**  Theoretically, setting the ratio to 0.5 maximizes the uncertainty of modality presence. This prevents the fusion module (LAMMA) from developing a bias or dependency on a specific "dominant" modality.
> - **Empirical Robustness:**  As shown in the table, setting the ratio to 0.5 yields the highest **AP@30** for the weaker Camera branch (80.81% vs. 72.76% at ratio 0.3). This indicates that an unbiased dropout strategy is crucial for activating the weaker branch to detect objects effectively, avoiding the "lazy" behavior where the model over-relies on LiDAR.
>
> **2. Why Overall Dropout Ratio**  **=**  **0.5? (Balancing Robustness vs. Peak Performance)**
>
> We further investigated the impact of the total probability of applying dropout (keeping LiDAR ratio fixed at 0.5).
>
> | Dropout Ratio | Best Val Epoch | L+C   | L      | C      |
> | ------ | -------- | --------- | ---------- | ---------- |
> | 0.3           | 31             | 98.42/98.10/95.28 | 96.86/96.64/94.38 | 73.63/67.30/47.82 |
> | 0.5           | 27             | 98.30/97.94/94.64 | 97.32/97.07/94.06 | 80.81/69.63/44.82 |
> | 0.7           | 25             | 98.06/97.59/94.02 | 97.48/97.17/94.42 | 79.29/70.97/48.84 |
> | 0.9           | 31             | 98.19/97.78/94.37 | 97.73/97.40/94.86 | 80.74/72.48/51.22 |
> | 1.0           | 29             | 97.79/97.31/93.43 | 97.55/97.21/94.58 | 82.57/72.93/50.75 |
>
> - **Trade-off Observation:**  There is a clear trade-off: increasing the dropout ratio improves Single-Modality performance (L/C) but slightly degrades Multi-Modal performance (L+C). For instance, Ratio 1.0 (training solely on single modalities) causes a noticeable drop in L+C AP@70 (93.43%) compared to Ratio 0.3 (95.28%), validating the necessity of retaining joint-modality samples.
> - **The "Sweet Spot":**  The ratio of 0.5 strikes the optimal balance. Compared to 0.3, it significantly boosts Camera AP@30 ( **+7.18%** ) with only a negligible drop in L+C AP@70 ( **-0.64%** ). Additionally, it achieves faster convergence (Best Epoch 27) compared to other settings.
>
> **Conclusion:** The choice of 0.5 for both ratios is empirically justified as the **Pareto-optimal** setting. It successfully awakens the independent capability of each modality without sacrificing the collaborative gain of feature fusion, confirming its effectiveness for the SiMO framework.
>
> > 2. While LAMMA fusion is designed to handle missing modalities, how does it differ from simpler alternatives—such as replacing the missing modality’s feature with a zero tensor in existing fusion schemes?
>
> Replacing missing features with zero tensors is a common heuristic, but it differs fundamentally from LAMMA in two key aspects:
>
> 1. **Implicit vs. Explicit Handling:**  Zero-padding is a "hard-switching" strategy that requires an external mechanism to detect failure and trigger the replacement. LAMMA handles missing data **implicitly and structurally** through the attention mechanism. The system does not need to "know" a sensor has failed; it simply attends to the available features.
> 2. **Granularity (Global vs. Local):**  Zero-replacement typically operates at the global feature map level (all or nothing). LAMMA, however, operates at the pixel level. As demonstrated in Figure 5, LAMMA can handle **partial/local failures** (e.g., LiDAR points missing in a specific sector) by adaptively attending to image features in those specific regions. A zero-tensor approach cannot address such heterogeneous, fine-grained failures.
>
>
> We are deeply grateful for your recognition of our work. We hope the additional experiments and explanations provided above address your concerns, and we would be delighted to elaborate further should you have any more questions.

---

### Author Response · Authors · 2025-11-30
**Closing Statement: Summary of Rebuttal and Acknowledgments**

The discussion phase has now concluded. We would like to take this opportunity to summarize our rebuttal efforts and express our sincere gratitude to the reviewers.

We are deeply grateful for the time and energy the reviewers dedicated to providing insightful comments, which have significantly helped us clarify our contributions and strengthen the reliability of our work. We carefully valued every piece of feedback and provided detailed point-by-point responses. We had hoped to receive further feedback from the reviewers to confirm whether our additional experiments and clarifications fully resolved their concerns. However, we regret that due to unexpected technical issues on the platform, some reviewers may have been unable to provide further evaluation or replies.

Regardless of these technical hurdles, we have made our utmost effort to address the questions raised. Below is a summary of our responses to the key concerns:

**Response to Reviewer Ntee:**

Regarding the **Dropout Ratio (RD)**, we conducted a comprehensive sensitivity analysis. The results empirically justified that setting the ratio to 0.5 strikes a "Pareto-optimal" balance, adhering to the maximum entropy principle to prevent modality bias while maintaining single-modality operability.


Regarding **LAMMA vs. Zero-padding**, we clarified that LAMMA offers an implicit, pixel-level attention mechanism that handles partial and heterogeneous failures, which structurally differs from the global "hard-switching" of zero-padding.


**Response to Reviewer AEwg:**


**Special Thanks:** We sincerely thank Reviewer AEwg for their constructive engagement and for acknowledging our clarifications by **raising the score**.

We addressed the concern regarding **stronger baselines** by implementing UniBEV with CNW fusion in the multi-agent setting. The results demonstrated that without our proposed alignment strategy, even robust single-agent methods suffer from modal branch collapse (Camera AP ≈ 0), confirming the necessity of SIMO.


We also provided additional experiments on **pose error robustness** and clarified that the performance limitations on **DAIR-V2X** stem from the dataset's lack of multi-view parallax rather than the model architecture.


**Response to Reviewer gwXL:**

We elucidated the unique challenge of **Heterogeneous Modal Failure** in Multi-Agent Collaborative Perception (MACP). We showed that unlike single-agent scenarios, MACP requires strict inter-agent semantic alignment to prevent failure during cross-agent fusion.

Regarding **training complexity**, we explained that our multi-stage pipeline is a deliberate design choice to prioritize training determinism and stability over the "hidden costs" and uncertainty often associated with joint training.

Furthermore, in response to the reviewer's requests, we have supplemented and clarified certain ambiguous sections of the paper. These revisions have been incorporated into the updated PDF.

**Response to Reviewer nBTy:**

Regarding **Training Complexity**, we responded by explaining that SiMO is based on a deterministic computational cost, ensuring strict modality independence, which is a deliberate architectural trade-off. Regarding the **DAIR-V2X** performance, we cited SOTA benchmarks to clarify that the issue is intrinsic to the dataset's single-view nature and not a failing of our method.

In response to the request for **Late Fusion metrics**, we provided new baseline comparisons. While peak performance is comparable due to dataset saturation, we highlighted that SiMO offers superior robustness at the feature level. In contrast, Late Fusion heavily relies on the quality of object proposals and fails if a modality completely misses the object.


**Conclusion**

Finally, we would like to extend our gratitude to all the reviewers and the Area Chair for their efforts during this review cycle. While the unexpected technical disruption at the end of the discussion period is regrettable, we believe that every challenge in the pursuit of truth is merely the darkness before the dawn. We remain committed to contributing to this community.

Sincerely,

The Authors

---

### Meta-Review · Area_Chair_ipdc · 2026-01-05

**Summary:**

This paper focuses on the realistic and safety-critical problem of modality failure in multi-agent collaborative perception. The problem formulation is clear, the motivation is well grounded, and the work is clearly differentiated from prior literature. The proposed SiMO framework, combining LAMMA-based structural fusion with the PAFR multi-stage training strategy, effectively mitigates modality competition and feature misalignment, and demonstrates strong and consistent robustness across multiple datasets and modality-failure scenarios.

While some reviewers raised concerns regarding training complexity, limited real-world validation, and the inherent limitations of monocular camera performance, these issues were largely addressed in the rebuttal through additional ablation studies, stronger baseline comparisons, robustness analyses (e.g., pose noise), and more carefully scoped claims. The authors also clarified the unique challenges of MACP compared to single-agent settings. Overall, the reviews trend positive, and considering the paper’s novelty, technical soundness, and quality of responses, the contribution is solid and acceptance is well justified.

**Reviewer Concerns:**

Addressed concerns:
The rebuttal effectively addressed most major technical concerns. The choice of RD dropout ratio was justified through detailed sensitivity analyses and theoretical reasoning. Questions about LAMMA versus zero-padding were clarified by highlighting its implicit, fine-grained attention-based handling of partial modality loss. Concerns on weak camera-only performance and lack of strong baselines were mitigated via additional comparisons (e.g., UniBEV) and a clearer framing of operability vs. absolute performance. Robustness to pose noise, feature alignment validity, and MACP-specific challenges were also supported by new experiments and analyses, leading some reviewers to raise their scores.

Outstanding concerns:
A few aspects remain open for future improvement, such as the relatively complex multi-stage training pipeline, the limited scope of real-world validation due to current dataset availability, and the absence of experiments with more than two modalities or very large agent populations. These points primarily reflect practical and data-related constraints rather than fundamental weaknesses of the proposed approach, and they do not detract from the soundness, generality, or core contributions of the work.

**Reviewer Scores:**

Overall, Reviewer AEwg would raise their score after the rebuttal because the authors provided comprehensive additional experiments, stronger baseline comparisons, and clearer positioning of SiMO’s claims, which directly addressed the reviewer’s main concerns regarding robustness, baselines, and practical relevance. Reviewer gwXL would maintain a strong accept-level score, as the paper was already viewed as technically solid and well motivated, and the rebuttal further reinforced this positive assessment with deeper analyses and clarifications. Reviewer Ntee and Reviewer nBTy are likely to keep their original scores, since their concerns about training complexity and real-world validation reflect broader practical limitations rather than clear flaws in the proposed method; nevertheless, the rebuttal substantially reduced uncertainty by justifying key design choices and clarifying scope. Taken together, these responses improve confidence in the method’s validity and positioning, leading to a slightly more favorable overall score distribution that supports an acceptance decision.

---

### Decision · Program_Chairs · 2026-01-26

Accept (Poster)